# Algebraic Recognition Approach in IoT Ecosystem

**Anvar Kabulov** [1,2] ![ID], **Islambek Saymanov** [1,2,3,*] ![ID], **Akbarjon Babadjanov** [4] and **Alimdzhan Babadzhanov** [5]

1   School of Mathematics and Natural Sciences, New Uzbekistan University, Mustaqillik Ave. 54, Tashkent 100007, Uzbekistan; kabulov_a@nuu.uz
2   Applied Mathematics and Intelligent Technologies Faculty, National University of Uzbekistan, Tashkent 100174, Uzbekistan
3   College of Engineering, Central Asian University, Milliy Bog' Street 264, Tashkent 111221, Uzbekistan
4   Department of Information Systems, University of Maryland, Baltimore County, 1000 Hilltop Circle, Baltimore, MD 21250, USA; baba4@umbc.edu
5   Department of Algorithmization, Engineering Federation of Uzbekistan, Tashkent 100003, Uzbekistan; a.babadjanov@uzmf.uz
*   Correspondence: saymanov_i@nuu.uz

**Abstract:** The solution to the problem of identifying objects in the IoT ecosystem of the Aral region is analyzed. The problem of constructing a correct algorithm with linear closure operators of a model for calculating estimates for identifying objects in the IoT ecosystem of the Aral region is considered. An algorithm operator is developed, which is considered correct for the problem $Z$, is the sum of $q$ operators of the assessment calculation model, and is described by a set of numerical parameters $3 \cdot n \cdot m \cdot q$, where $n$ is the number of specified features, $m$ is the number of reference objects, and $q$ is the set of recognized objects. Within the framework of the algebraic approach, several variants of linear combinations of recognition operators are constructed, the use of which gives the correct answer on the control material, and this is proven in the form of theorems. The constructed correct recognition algorithms, which are the easiest to use, where there is no optimization procedure, make it possible to quickly solve the issue of identifying incoming information flows in the IoT ecosystem of the Aral region.

**Keywords:** algebraic recognition; correct algorithm; ecosystem; IoT; analyzing; operation; associative; commutative; distributive

**MSC:** 68T10

## 1. Introduction

In recent years, solutions to applied problems of classification recognition and prediction have achieved great developments. In many real cases, the solution scheme remains the same; the set of possible solutions is divided into subsets in such a way that solutions that are close in some metric fall into one subset [1,2]. In the future, solutions that fall into the same subset do not differ, and all objects corresponding to these solutions are included in one class [3]. The information gained from past experience is presented in the following form: various objects are described in some way and their descriptions are divided into a finite number of non-overlapping classes [4]. When a new object appears, a decision is made to assign it to one class or another [5]. It is proposed to choose a generalized algorithm such that it achieves extreme forecasting quality [6]. Let us consider algorithmic models for solving classification problems. Among these models, we can identify the most frequently encountered ones when solving applied problems.

The work [7] discusses practical precedent-based recognition algorithms based on logical or algebraic correction of various heuristic recognition algorithms. The recognition problem is solved in two stages. First, algorithms of a certain group are applied independently to recognize an arbitrary object, then an appropriate corrector is applied to calculate

the final collective solution. The general concepts of the algebraic approach, descriptions of practical algorithms for logical and algebraic correction, and the results of their practical comparison are given.

In [8], the problem of classification (recognition) based on precedents was studied. The issues of increasing the recognition ability and learning speed of logical correctors—recognition procedures based on constructing correct sets of elementary classifiers—have been studied. The concept of a correct set of elementary classifiers of a general form is introduced, and on this basis, a qualitatively new model of a logical corrector is constructed and studied. This model uses a wider class of correcting functions than previously constructed models of logical correctors.

In ref. [9], the supervised classification problem with a large number of classes was studied and the ECOS circuit (error output codes) was optimized. First, an initial binary matrix was randomly formed, the number of rows was equal to the number of classes, and each of the columns corresponded to the union of several classes into two macroclasses. In the ECOS approach, the binary classification problem was solved for the recognized object and each association. The object belonged to the class whose code string was closest. A generalization of the ESOS approach was given, offering a solution to the discrete optimization problem when searching for optimal associations, the use of the probabilities of correct classification in dichotomous problems, and the degree of information content of dichotomies. If the algorithms for solving dichotomous problems are correct, then the algorithm for recognizing the original problem is also correct.

In ref. [10], the complexity of the logical analysis of integer data was studied. For special problems of searching for frequent and infrequent elements in data, on the solution of which the training of logical classification procedures is based, the asymptotics of the typical number of solutions were given. The technical basis for obtaining these estimates was the methods for obtaining similar estimates for the intractable discrete problem of constructing (enumerating) dead-end coverings of an integer matrix, formulated in the work as the problem of finding "minimal" infrequent elements. The new results mainly concern the study of metric (quantitative) properties of frequently occurring elements.

The purpose of the research in this article is to develop effective object recognition methods to solve the identification problem in the IoT ecosystem of the Aral region. The objective of the research is to develop correct recognition algorithms based on an algebraic approach in solving the problem of identifying [11–15] the flow of information in the IoT ecosystem of the Aral region, which is described in detail in work [15] of the authors of this paper.

The IoT ecosystem of the Aral region covers a network of sensors for determining groundwater levels and salinity of water and soil and transmitting information via communication channels to the system server for identification and further processing. This article solves the applied problem of identification in the ecosystem of the Aral region, where formulas for recognizing correct algorithms are specifically given. Moreover, the correctness is proven in the theorems of this work in terms of the algebraic approach.

The public benefit of the results of this study is that the results are used in the ecosystem of the Aral region in monitoring the use of water resources and calculating salinity levels for agricultural use in an interactive mode.

The problems of recognizing disjoint classes are considered in refs. [8,16–18]. The methodology used to develop and test the algebraic recognition approach was proposed and described in the works of academician Yu.I. Zhuravlev and his students. In this work, based on the idea of the algebraic approach of academician Yu.I. Zhuravlev, a much simpler and significantly more efficient computational algorithm of the nearest neighbor and an average distance algorithm are proposed for solving these problems using algebraic methods. Methods have been developed that make it possible to more economically encode the recognition operator, which reduces the required memory and more efficiently uses the constructed correct algorithms for solving applied problems. The algebraic approach shows that each algorithm $A$ can be represented as $A = B \times C$, two operators, recognition

operator $B$, and the decision rule $C$. Within the framework of the algebraic approach, several variants of linear combinations of recognition operators $A$ are constructed, the use of which gives the correct answer on the control material, and this is proven in the form of theorems.

Thus, taking into account the analysis of the works of scientists in the field of recognition researched on the topic of the article, this work investigates the problem of constructing a correct algorithm with linear closure operators for a model for calculating estimates. An algorithm operator was developed, which is considered correct for problem $Z$, represents the sum of q operators from the model for calculating estimates, and is described by a set of $3 \cdot n \cdot m \cdot q$ (where $n$ is the number of predetermined features, $m$ is the number of reference objects, $q$ is the set of recognized objects) numerical parameters. An operator belonging to the linear closure of a model of the type of calculation of estimates was constructed [19–22]. The completeness of the linear closure of this model was proven for all problems in which for each class there is at least one stationary pair $(u, v)$, and this correct algorithm is written explicitly.

## 2. Models Based on the Calculation of Estimates

In the work [23], the so-called parametric recognition algorithms were considered, such collections of algorithms in which each algorithm is encoded in a one-to-one way by a set of numerical parameters. In these models, the proximity between parts of previously classified objects and the object to be classified was analyzed [24]. Based on a set of estimates, a general estimate for the object was developed and, according to the introduced decision rule, the belonging of the recognized object to one or another class was determined [25,26].

In the article, as the initial model ($A$), a model was considered that is related to the model for calculating estimates, supplemented by some simple type recognition algorithms: the nearest neighbor algorithm, the average distance algorithm, etc.

A feature of the algorithms of this class is that for calculating estimates that determine the belonging of a recognized object, there are simple analytical formulas that replace complex enumeration procedures that arise when calculating proximity estimates using a system of support sets [27].

In these models, the division of the algorithm into recognition operators and decision rules was carried out in a natural way [28–32].

We will consider only algorithms represented in the form $A = B \cdot C$, where $B$ is an arbitrary recognition operator. It turns out that an essential part of the algorithm is the operator—$B$; decision rule—$C$ can be made standard for all algorithms and programs. Any recognizing vote operator maps task $Z$ to a numeric matrix of votes or scores $B(Z) = \left\| G_{ij} \right\|_{q \cdot l}$, $G_{ij} = G_j(S^i)$; moreover, the value $G_{ij}$ has a clear, meaningful interpretation. This value can be considered as the degree of belonging of the examined object $S^i$ to the class expressed by a number $K_j$. After introducing appropriate normalizations, the value $G_{ij}$ can also be considered as the value of the membership function of elements $S^i$ of the set $K_j$.

Before introducing the submodel used in what follows, let us write how the algebra of recognizing operators is constructed [33,34]. Let $Z = (J, \tilde{S}^2)$ be a fixed recognition problem with classes $K_1, \ldots, K_l$, also let $B_1$ and $B_2$ recognize operators, $c$ is a real number

$$B_i(Z) = \left\| G_{uv}^i \right\|_{q \cdot l}, \ i = 1, 2. \tag{1}$$

Then, the sum and product of the operators $B_1$ and $B_2$ as well as the multiplication of the operator by a real number is defined as follows

$$(B_1 + B_2)(Z) = \left\| G_{uv}^1 + G_{uv}^2 \right\|_{q \times l} \tag{2}$$

$$(B_1 \times B_2)(Z) = \left\| G_{uv}^1 \times G_{uv}^2 \right\|_{q \times l} \tag{3}$$

$$(cB)(Z) = \|c \cdot G\|_{q \times l}, \quad i = 1, 2. \tag{4}$$

Obviously, all these operations are commutative and associative; moreover, the operation of addition is distributive with respect to the operation of multiplication by a number. Due to these properties, if $(B)$ is the original set of operators, $a(B)$ is the closure of the family $(B)$ with respect to the introduced operations (algebraic closure) from $a(B)$ can be represented as operator polynomials $\sum b_{i1}, \ldots, b_{ik}, B_{i1} \cdot B_{i2} \cdot \ldots \cdot B_{ik}$.

Here, $b_{ik}$—constants and original operators $B_{ik}$—play the role of variables for ordinary polynomials.

Note that if we have one algorithm for applying the original operators $B_{ik}$ to any problem $Z$, then it is easy to construct an algorithm for applying the operator polynomial to $Z$.

For any operator polynomial, the algorithm to be applied to the problem is constructed in a similar way: the original operators are applied to the problem, and then the resulting matrices are multiplied by the corresponding scalars and finally added, as in the example just considered. The evaluation matrix obtained by applying the algorithm $a(B)$ from the closure algebra does not allow such a simple interpretation as the matrix of the number of votes in voting algorithms [16–18]. The new recognizing operators are a formal extension of the original space of meaningful operators. Such formal extensions are often used in mathematics; thus, the field of complex numbers is a formal extension of the field of real numbers, for example, the process of formal Galois extensions is known. In the Galois extension, an algebraic equation of any degree is solved elementarily. However, for a long time, no physical interpretation of the Galois expansion was found [35]. Only in recent years has it been established that the elements of the Galois extension are naturally interpreted in connection with problems of error correction in information transmission.

There is currently no meaningful interpretation for the algebraic extension of the space of operators. However, with the help of these extensions, difficult extremal problems are relatively easily solved, including the problem of synthesizing an error-free algorithm for a given recognition problem [36].

The degree of an operator polynomial is introduced similarly to the degree of an ordinary polynomial in many variables from the terms:

$$b_{i1}, \ldots, b_{in} \quad B_{i1} \cdot B_{i2} \cdot \ldots \cdot B_{ik} \tag{5}$$

the term with the largest $K$ equal, for example, is chosen, and the degree of the polynomial is assumed to be equal to $K$. Based on the definition of the degree of polynomials, it is easy to distinguish in the extension $a(B)$ a system of nested extensions

$$a^0\{B\} = \{B\}, a^1\{B\} = \left\{\sum b_i\{B_i\}\right\}, \ldots, a^k\{B\} = \left\{ \begin{array}{c} \text{the totality of all} \\ \text{polynomials of degree} \\ \text{not higher than K} \end{array} \right\}$$

obviously, $a^0\{B\} \leq a^1\{B\} \leq \ldots \leq a^k\{B\} \ldots a\{B\}$ the set $a^k\{B\}$ is called the $k$-th power extension of the original space of operators $\{B\}$. The set is of particular importance $a^1\{B\}$, it consists of all possible linear forms from the original operators $\{B\}$. The elements are represented as:

$$\sum b_i B_i, \quad B_i \in \{B\} \tag{6}$$

When forming the elements included in the $a^1\{B\}$ operation, the product of operators is not used [37,38]. Therefore, this extension is an extension of the original space using the "+" operation, "multiplication by a number"; therefore, the set is $a^1\{B\}$, also commonly denoted as $L\{B\}$ and is called a linear extension of the set $\{B\}$. And so, let a set of $\{A\}$ algorithms be given $\{A\}$, and each $A$ is represented as $A = B \times C$ and $\{B\}$ the initial model of operators. We take a fixed decision threshold rule $C(d_1, d_2)$. We introduce a family of algorithms $L\{B\} \cdot C(d_1, d_2), \ldots, a^k\{B\} \cdot C(d_1, d_2), \ldots, a\{B\} \cdot C(d_1, d_2)$ called, respectively, a linear extension $L\{A\}$, an algebraic extension of the $k$-th degree $a^k\{A\}$, and an algebraic extension $a\{A\}$ of a family of algorithms $\{A\}$.

We see that the constructed sets of algorithms consist, in the execution of the corresponding operators, and a fixed threshold decision rule c is applied to the result of the operator's action $c(d_1, d_2)$.

Let some set of recognition problems be given $\{Z\}$ and the initial set of recognizing operators be chosen in some way $\{B\}$. Let also within the framework $\{B\}$, not for every problem $\{B\}$, there exist an operator $B_Z$ and an algorithm $B_Z \times C_Z$, which gives an error-free solution to the problem $Z$. Then, the scheme for constructing a correct recognition algorithm consists of the following steps:

Stage 1. Some extension is chosen $a^k\{A\}$ in which the existence of an error-free solution is guaranteed for each problem $Z$ from $\{Z\}$. In this case, it is natural to find, if possible, the smallest degree of expansion of $K$ [39]. The implementation of the first stage in the literature is usually called the study of the completeness of the expansion. The theorems proved in this case are usually analogous to the existence theorem.

Stage 2. In the chosen extension, $a^k\{A\}$ for an admissible problem $Z$, either an error-free (correct algorithm) is constructed, or, if the former is associated with large computational difficulties, an algorithm for solving with respect to, which is sufficiently acceptable in accuracy $Z$. The exact calculation of the minimum degree of $K$ is a laborious task, which at present can be solved only for relatively narrow classes of problems $\{Z\}$, and the original family of algorithms $\{A\}$. Therefore, in the studies carried out for the minimum degree of expansion, an upper estimate is constructed that guarantees the completeness of the expansion. So, for the voting model, such an estimate is

$$K = \left\lceil \frac{\ln q + \ln l + \ln(d_1 + d_2) + \ln d_2 - \ln d_1}{\ln\left(1 - \frac{1}{q}\right)} \right\rceil \tag{7}$$

built in [16].

Here, $q$ is the number of recognizable objects, $l$ is the number of classes, $d_1, d_2$ and are the parameters of the threshold decision rule.

For most real problems, this estimate is overestimated, and therefore, the algorithm built in the second stage has computational redundancy. Of particular importance is the fact that the above estimate is constructed for problems with intersecting classes, while the majority of real problems are problems with non-intersecting classes. Later, we will see that for a wide class of problems it is sufficient to consider the degree of 1, that is, to use only a linear closure.

Algorithms of the class for calculating estimates allow solving recognition problems of all types: assigning an object to one of the given classes, automatic classification, choosing a feature system to describe recognition objects, and evaluating their effectiveness.

## 3. Algebraic Methods for Solving Recognition Problems with Non-Crossing Classes

In [12], algebraic methods for solving recognition problems with a finite number of intersecting classes were developed. For each recognition problem $Z$ in terms of algebra over families of heuristic algorithms, a correct algorithm was constructed, e.g., an algorithm that correctly classifies a given finite sample of objects for each of the classes. Due to the fact that the problem with intersecting classes was considered, the algorithm constructed in the above works is rather cumbersome [40–45]. The description of the algorithm itself requires the use of large memory (the amount of memory grows proportionally) $q^2 \cdot l^2$, where $q$—the number of recognition objects in a given task, $l$—the number of classes. It turns out that if we consider only problems with non-intersecting classes, then similar algebraic methods can be used to obtain a much simpler description and a much more effective computational algorithm.

Basic concepts and designations. Let a set of admissible objects be given $\{S\}$ and it is known that $\{S\}$ can be represented as the sum of a finite number of disjoint subsets $K_j, j = \overline{1,l}$ called classes

$$\{S\} = K_1 \bigcup \ldots \bigcup K_l, K_u \bigcap K_v = \varnothing, u = \overline{1,l}, v = 1, 2, \ldots, u-1, u+1, \ldots, l.$$

Objects $S$ are descriptions of some real objects using successive values of a finite number of predefined features $1, 2, 3, \ldots, n$. Each attribute i can be associated with its set of values $M_i$, which we will assume to be a metric space with distance $\rho_i(xy)$. This paper considers the following sets $M_1, \ldots, M_n$ and metrics $\rho_1, \ldots, \rho_n$.

1. $M_i$—there is a finite or infinite interval, a half-interval, or a finite segment of the numerical axis. Then, $\rho_i(xy) = |x - y|$. In this case, the sign i is called numerical.

2. $M_i = \{0, 1\}$ then $\rho_i(xy) = |x - y|$ the sign $i$ is called binary.

3. $M_i = 0, 1, \ldots, k_i - 1$—a finite set of integers; then defined by the following Table 1:

**Table 1.** Set of integers.

| $Y$ \ $X$ | 0 | 1 | 2 | ... | ... | ... | $K_{i-1}$ |
|---|---|---|---|---|---|---|---|
| 0 | 0 | $\rho_{01}$ | $\rho_{02}$ | ... | ... | ... | $\rho_{0(k_i-1)}$ |
| 1 | $\rho_{10}$ | 0 | $\rho_{12}$ | ... | ... | ... | $\rho_{1(k_i-1)}$ |
| 2 | $\rho_{20}$ | $\rho_{21}$ | 0 | ... | ... | ... | $\rho_{2(k_i-1)}$ |
| ... | ... | ... | ... | ... | ... | ... | ... |
| ... | ... | ... | ... | ... | ... | ... | ... |
| ... | ... | ... | ... | ... | ... | ... | ... |
| $k_i - 1$ | $\rho_{k_i-1,0}$ | $\rho_{k_i-1,1}$ | ... | ... | ... | ... | 0 |

There are zeros on the diagonal in the table, in addition (the table is symmetrical with respect to the main diagonal).

Not every table of the specified type defines a metric. A necessary and sufficient condition for the latter is the fulfillment of inequalities for any $u, v, t$ such that $0 \leq u, v, t \leq k - 1$ occurs $\rho(u, v) = t$.

These inequalities ensure that the triangle axiom holds true. Signs are called graded or scalable.

4. $M_i = \{a_{i1}, \ldots, a_{ik}(i)\}$ finite non-numeric set.

Then $\rho(x, y) = \begin{cases} 1, & \text{if } x \neq y \\ 0, & \text{if } x = y \end{cases}$.

The attribute in this case is called named.

In the future, we will consider only numerical, binary, graded, and named features.

A set $\{S\}$ is a collection of sets; $(a_1(S), \ldots, a_n(S))$, such cases will be noted separately.

Consider the predicates $P_i(S) = S \in K_j, j = \overline{1, l}$. It is easy to notice that for each $S$, one of these predicates is equal to 1 and the rest are equal to 0.

The task of recognizing $Z$ is to use some information $J_0$ about the sets $\{S\}, K_1, \ldots, K_l$ compute the value of the predicates $P_j(S)$, for each finite number of objects $S^1, \ldots, S^q \in \{S\}$. In view of the previous remark, this is the same as specifying for each $S^i \in \tilde{S}^q = \{S^1, \ldots, S^q\}$ the number of the predicates $t$ such that $P_t\{S^i\} = 1$. The latter distinguishes the recognition problem with non-overlapping classes from the general recognition problem.

For a complete formalization of the description of task $Z$, it is necessary to determine the initial information $J_0$.

In this paper, we restrict ourselves to only one type: $J_0$—consists of an enumeration of reference objects, for each of which the number of the class containing this object is indicated.

For non-overlapping classes, information $J_0$ can also be in the form of a learning table $T_{nml}$, where $n$—determines the number of features, $m$—the number of objects, $l$—the number of classes:

$$J_o = T_{nml} = \begin{pmatrix} a_{11} & a_{12} & \ldots & a_{1n} & \\ \ldots & \ldots & \ldots & \ldots & K_1 \\ a_{m_1 1} & a_{m_1 2} & \ldots & a_{m_1 n} & \\ \ldots & \ldots & \ldots & \ldots & \ldots \\ a_{m_{j-1} 1} & a_{m_{j-1} 2} & \ldots & a_{m_{j-1} n} & \\ \ldots & \ldots & \ldots & \ldots & K_j \\ a_{m_j 1} & a_{m_j 2} & \ldots & a_{m_j n} & \\ \ldots & \ldots & \ldots & \ldots & \ldots \\ a_{m_{l-1},1} & a_{m_{l-1},2} & \ldots & a_{m_{l-1},n} & \\ \ldots & \ldots & \ldots & \ldots & K_l \\ a_{m1} & a_{m2} & \ldots & a_{mn} & \end{pmatrix} \qquad (8)$$

We will use such standard information presented in the form of a table. We will always count:

$$S_1, \ldots, S_{m_1} \text{—belongs to } K_i \qquad (9)$$

$$S_{m_{j-1}}, \ldots, S_{m_j} \text{—belongs to } K_j \qquad (10)$$

$$S_{m_{l-1}+1}, \ldots, S_m \text{—belongs to } K_l \qquad (11)$$

We introduce important notation for what follows:

$$\left\{ S_{m_{j-1}}, \ldots, S_{m_j} \right\} = \tilde{K}_j, \quad \{S_1, \ldots, S_m\} / \tilde{K}_j = C\tilde{K}_j, \quad j = 1, 2, \ldots, l, \quad m_0 = 0.$$

The set $\tilde{K}_j$ consists of all reference objects belonging to the class $K_j$.

In recognition problems with intersecting classes, algorithms $A$ were considered such that $A(Z) = A\{J_0, \tilde{S}^q\} = \|x_{ij}\|$ where $x_{ij} = P_i(S^i)$.

These algorithms calculate information vectors for each object

$$\tilde{\alpha}\left(S^i\right) = (\alpha_{i1}, \alpha_{i2}, \ldots, \alpha_{il}) = \left( P_1\left(S^i\right), P_2\left(S^i\right), \ldots, P_l\left(S^i\right) \right)$$

and are called correct for problem $Z$.

For problems with intersecting classes, arbitrary binary vectors can be used as information vectors.

For problems with non-intersecting classes, only those containing exactly one-unit coordinate are informational, the rest of the coordinates are equal to zero. The last remark will be essential in what follows.

Let $A$ be an arbitrary algorithm that translates the recognition problem $Z$, with $l$—classes, $Z = \{J_0, \tilde{S}^q\}$ into the matrix of answers $\|\beta_{ij}\|_{q \times l}$, $A\left(J_0, \tilde{S}^q\right) = \|\beta_{ij}\|_{q \times l}$, $\beta_{ij} \in \{0, 1, \Delta\}$.

Equality $\beta_{ij} = 1, \beta_{ij} = 0, \beta_{ij} = \Delta$ means, respectively, that the algorithm $A$ calculated for the object $S^i : S^i \in K_j, S^i \notin K_j$, turns out to be from the calculation of the object's belonging $S^i$ to the class $K_j$. If $\beta_{ij} \in \{0, 1\}$, then this also does not mean that, that $\beta_{ij} = P_j(S^i)$ is, the algorithm can also make errors in addition to failures.

Such algorithms are called incorrect for problem $Z$. Obviously, correct algorithms are a special case of incorrect ones.

For arbitrary incorrect algorithms (and hence for correct ones) hold.

**Theorem 1.** *Each algorithm $A$ can be represented as $A = B \times C$ (multiplication means sequential execution), and if $A(Z) = \|\beta_{ij}\|_{q \times l}$, $\beta_{ij} \in \{0, 1, \Delta\}$ that $B(Z) = \|a_{ij}\|$—numerical matrix,*
$$C\left(\|a_{ij}\|_{q \cdot l}\right) = \|\beta_{ij}\|_{q \cdot l}.$$

Theorem 1 [16] shows that each algorithm $A$ can be divided into two successive stages. In the 1st stage, task $Z$ is converted into a numerical matrix of standard sizes $q$—rows,

*l*—columns, the number of rows is equal to the number of recognized objects, and the number of columns is equal to the number of classes.

In the 2nd stage, according to this numerical matrix, answers are finally formed to questions about the belonging of objects $S^i, \ldots, S^q$ to classes $K_1, \ldots, K_l$.

The value $a_{ij}$ is naturally interpreted as the values of the measures of belonging of objects $S^i$ to classes $K_j$. Stage *B* is called the recognition operator, stage *C* is the decision rule.

In what follows, only threshold decision rules are considered.

$$C^* \left( \|a_{ij}\|_{q \cdot l} \right) = \|C(a_{ij})\|_{q \cdot l} \tag{12}$$

The rule is applied element by element. Let *a* be a number and $d_1, d_2$ also be numbers (thresholds) and $0 < d_1 < d_2$, then

$$C^*(a) = \begin{cases} 1, & \text{if} \quad a > d_2 \\ 0, & \text{if} \quad a < d_1 \\ \Delta, & \text{if} \quad d_1 < a < d_2 \end{cases} \tag{13}$$

## 4. Basic Model of Recognizers (*B*)

*Simple Heuristic Operators*

Let, according to the accepted notation, the initial information $J_0$ presented in the form of a table $T_{nml}$ (see Section 3).

Also let the set of features $1, 2, \ldots, n$ be divided into subsets

$$\hat{M}_1 = \{1, 2, \ldots, n_1\}, \ \hat{M}_2 = \{n_1 + 1, \ldots, n_2\}, \ \hat{M}_3 = \{n_2 + 1, \ldots, n_3\}, \hat{M}_4 = \{n_3 + 1, \ldots, n\}$$

numerical, binary, graded, and named features.

For each of the signs, a metric $\hat{M}_1$ is introduced $\rho_i(x, y) = |x - y|$.

For each of the features included in $\hat{M}_2, \hat{M}_4$, a metric is introduced

$$\rho_i(x, y) = \begin{cases} 1, & x \neq y \\ 0, & x = y \end{cases} \tag{14}$$

For each of the graded features included in $M_3$, either a metric is specified $\rho_i(x, y) = |x - y|$, or a metric is defined using a table specifying $(x, y)$ values for each pair $\rho_i(x, y)$ (See the definition of the metric in Section 3).

Let $S = (a_1, \ldots, a_n), S' = (b_1, \ldots, b_n)$ admissible objects.

Let us put

$$\rho_1(S, S') = \min_{i=\overline{1,n}} \rho_i(a_i, b_i) \tag{15}$$

$$\rho_2(S, S') = \frac{1}{n} \sqrt{\sum_{i=1}^{n} \rho_i^2(a_i, b_i)} \tag{16}$$

$$\rho_3^{\alpha\beta}(S, S') = \alpha \cdot \rho_1(S, S') + \beta \cdot \rho_2(S, S') \qquad \alpha + \beta = 1, \ 0 \leq \alpha \leq 1 \tag{17}$$

We introduced two fixed metrics $\rho_1, \rho_2$ and a family of metrics $\rho_3^{\alpha\beta}$ depending on the parameter in the space of admissible objects $\alpha$. Any of the introduced metrics can be used in the operator described below.

The nearest neighbor operator $\beta_{\min}^1$ is as follows:

(a) in the selected metric $\rho$ of the three entered, the following are calculated $\rho(S, S') = \rho(t), t = m_{j-1} + 1, \ldots, m_j$ distance from the recognized object *S* to the objects included in the set $\tilde{K}_j$.

(b) The value is calculated $\min_t \rho(t) = p_j(S), j = 1, 2, \ldots, l, \ t = m_{j-1} + 1, \ldots, m_j$.

(c) The values are formed in the following way:

$$G_j(S) = \frac{1}{p_i(S)+1}, j = 1, 2, \ldots, l, \ B^1_{\min}(S) = (G_1(S), \ldots, G_j(S), \ldots, G_l(S)) \tag{18}$$

The greater the value, the closer in the selected metric is the recognizable object to the object $G_j(S)$ closest to it from, $\tilde{K}_j$, to the object included in the learning table and belonging to the class $K_j$. Obviously, if the operator $B^1_{\min}$ is consistently presented with a set of recognizable objects $S^1, \ldots, S^q$, then they will translate it into a matrix $\left\| G_j(S^i) \right\|_{q\cdot l}$.

Thus

$$B^1_{\min}(T_{nml}, \tilde{S}^q) = B^1_{\min}(Z) = \begin{Vmatrix} G_1(S') & \ldots & G_j(S') & \ldots & G_l(S') \\ \ldots & \ldots & \ldots & \ldots & \ldots \\ G_1(S^i) & \ldots & G_j(S^i) & \ldots & G_l(S^i) \\ \ldots & \ldots & \ldots & \ldots & \ldots \\ G_1(S^q) & \ldots & G_j(S^q) & \ldots & G_l(S^q) \end{Vmatrix} \tag{19}$$

The system of operators described by us $B^1_{\min}$ is somewhat different from the operators that are commonly called nearest neighbor operators. However, we have retained the operator's schematic diagram.

## 5. Average Distance Operator $B^1_{sr}$

Clause

(a) of the definition of this operator, exactly repeats the corresponding clause of the definition of the operator $B^1_{\min}$.

(b) The value is calculated $\frac{1}{m_j - m_{j-1}} \sum\limits_{t=m_{j-1}+1}^{m_j} \rho(t) = \rho_j(S)$

(c) as in the definition, $B^1_{\min}$ it is calculated $G_j(S^i) = \frac{1}{\rho_j(S)+1}$ $j = 1, 2, \ldots, l$

As for the operator $B^1_{\min}$, it is easy to see that

$$B^1_{sr}(Z) = B^1_{sr}(T_{nml}, \tilde{S}^q) = \begin{Vmatrix} G_1(S') & \ldots & G_j(S') & \ldots & G_l(S') \\ \ldots & \ldots & \ldots & \ldots & \ldots \\ G_1(S^i) & \ldots & G_j(S^i) & \ldots & G_l(S^i) \\ \ldots & \ldots & \ldots & \ldots & \ldots \\ G_1(S^q) & \ldots & G_j(S^q) & \ldots & G_l(S^q) \end{Vmatrix} \tag{20}$$

*Voting Type Operators*

Let the objects in the training information $K_j$ belong to the class $S_{u1}, \ldots, S_{ut}$, those $K_j = \{S_{u1}, \ldots, S_{ut}\}$.

In our notation $S_{ui} = \{a_{ui_1}, \ldots, a_{ui_n}\}$.

As before, $\rho_1, \ldots, \rho_n$ we denote the metrics in the sets $M_1, \ldots, M_n$ of feature values $1, 2, \ldots, n$. Let us enter the parameters: $\varepsilon_{rv} \geq 0, r = u_1, \ldots, u_t, v = 1, 2, \ldots, n$.

These parameters will further define the proximity function for the recognized object and the set $\tilde{K}_j$. Parameters are also entered $p_{rv} > 0, r = u_1, \ldots, u_t, v = 1, 2, \ldots, n$.

The parameter $p_{rv}$ is the weight of the $v$-th feature in the reference object $S_r$. There are no other parameters. The operator is defined as follows. In the set of pairs, $(r, v) : r = u_1, \ldots, u_t, v = 1, \ldots, n$ a subset is distinguished $\Omega$, which is called the support set of operators in what follows.

If the recognized object is equal to $S = (a_1, \ldots, a_n)$, then the proximity function $B(\Omega, S, \tilde{K}_j)$ is introduced as follows: if the $(r, v) \in \Omega$ inequalities $\rho_v(a_{rv}, a_{tv}) \leq \varepsilon_{rv}$ that $B(\Omega, S, \tilde{K}_j) = 1$.

Otherwise $B(\Omega, S, \tilde{K}_j) = 0$.

In other words, the proximity function is equal to 0 if $\Omega$ there is a pair $(r, v)$ in such that $\rho_v(a_{rv}, a_{tv}) > \varepsilon_{rv}$. The number of votes $G_j(S)$ for $S$ is calculated in the following way:

$$G_j(S) = \left( \sum_{(r,v) \in \Omega} p_{rv} \cdot B(\Omega, S, \tilde{K}_j) \right).$$

Obviously, if the proximity function $B$ for a set $\Omega$ is 0, then the number of votes in this set for a recognized object is 0. If $B = 1$, then the number of votes is equal to
$$G_j(S) = \sum_{(r,v) \in \Omega} p_{rv}.$$

In this case, the number of votes is equal to the sum of the weights of pairs $(r, v)$—a sign-reference for all pairs included in $\Omega$.

If a problem with non-intersecting classes is considered, then the sets $\tilde{K}_j, j = 1, 2, \ldots, n$ in the learning information do not intersect, and therefore, the parameters $\varepsilon_{rv}, p_{rv}$ are determined independently for each class $K$.

Thus, we have completely defined the operator that translates the training information and the object $S$ into a numeric string $(G_1(S), \ldots, G_l(S))$.

Similarly, the set of recognizable objects $S^1, \ldots, S^q$ is translated by the operator into a numerical matrix of voices $\left\| G_j\left(S^i\right) \right\|_{q \cdot l}$.

For problems with intersecting classes, in place of the parameters $\varepsilon_{rv}, p_{rv}$, one should consider $\varepsilon_{rv}^d, p_{rv}^d$ and enter them independently for each class. Thus, different parameters can be assigned to the same pair $(r, v), S_r \in J_0(n, m, l), 1 \le v \le n$ different parameters can be compared $\varepsilon_{rv}^1, \ldots, \varepsilon_{rv}^l, p_{rv}^1, \ldots, {}_{rv}^l$. But $G_j(S)$ only $S_{rv}^d, p_{rv}^d$.

The constructed voting model is an essential generalization of the basic model. Indeed, in order to obtain a basic model from it, it is necessary to set the parameter $\varepsilon_{rv}$ equal for different and identical $v$, and the parameters $p_{rv} = p_r \gamma_v$.

The fact that in this model one support set is considered, in contrast to the basic model, is not a limitation, since sums of operators can be considered. One can obtain an operator operating on an arbitrary system of support sets.

We will mainly explore this voting model. Let us show that already in a linear closure, under the fulfillment of natural assumptions, it is possible to construct an algorithm that is correct for any preassigned control sample.

The construction of the operator of this algorithm is very simple, and, accordingly, the algorithm, with a sufficiently large volume of training sample, gives the correct answer almost everywhere. In real calculations, it is not necessary to remember all the parameters, $p_{rv} \cdot \varepsilon_{rv}$ it is enough to limit ourselves to only a small part.

## 6. Completeness of Linear Closure of the Second Model Voting

We will consider problems with non-overlapping classes $K_1, \ldots, K_l$ and assume that the information $J_0$ is given in the form of a learning table $T_{nml}$. The task of recognition will be to classify the final sample $S^1, S^2, \ldots, S^q, \ S^i = (b_{i1}, b_{in})$.

As before, in what follows, we will assume that in the recognizable sample $\tilde{K}_j$ the objects belong to the class $S^{q_{j-1}+1}, \ldots, S^{q_j}, q = 0, q_l = q$.

The purpose of this section is to single out a set of basic operators of the considered model for calculating estimates, construct their linear closure, and prove the fact that for each class object $\tilde{K}_j$ from the sample these operators $\tilde{S}^q$ form a sufficiently $G_j$ large estimate $G_u$ and for $u \ne j$.

All basic operators, as well as operators from the linear closure, will be constructed explicitly.

We will first need a standard condition relating the type of training information $T_{nml}$ and a recognizing sample, $\tilde{S}^q$ namely, we will assume that:

for any two different objects $S^u, S^v$ from the collection $\tilde{S}^q$ among the objects included in $T_{nml}$ and belonging to the class $K$, there is such an object $S_t$ and such a sign, $r$ what $\rho_r(a_{tr}, b_{ur}) \ne \rho(a_{tr}, b_{ur})$.

In this case, it is customary to say that objects from the system $\tilde{S}^q$ are pairwise non-isomorphic.

When proving the completeness of a linear closure, we will rely on the notion of a marked pair. Since in this paper only problems with non-overlapping classes are considered, the notion of a marked pair will be somewhat changed.

**Definition 1.** *Pair* $(S^u, j), S^u \in \tilde{S}^q, \tilde{S}^u \in K_j, 1 \leq j \leq l$ *is called marked in the operator B if* $B_v(Z) = \|a_{tv}\|_{q \cdot l}, 1 \leq v \leq w, a_{uj} \geq 1, |a_{rw}| < \delta \to 0$ *for all S such that* $\tilde{S}^u \in K_r, r = 1, 2, \ldots, l.$

From the definition of a marked pair, it can be seen that such a pair appears in the operator *B* if: $a_{uj} = G_j(S^u)$ the estimate for an object $S^u$ by class $K_j$ is large enough, all estimates $a_{tv} = G_v(S^t)$ for the case when the object $S^t$ does not belong to the class $K_v$ are sufficiently small in absolute value.

Let $B_1, \ldots, B_w$ be given in some model such that each pair $(S^u, j), S^u \in K_j$ is marked with at least one operator $B_1$.

**Theorem 2.** *There is a linear combination* $\sum_{i=1}^{n} a_i B_i = \tilde{B} \in \tilde{L}\{B\}$ *such that* $\tilde{A} = \tilde{B} \cdot C$, *C the threshold decision rule algorithm is correct for problem Z.*

**Proof of Theorem 2.** Recall that the threshold decision rule *C* is defined by the relation $C\left(\|a_{tv}\|_{q \cdot l}\right) = \|C(a_{tv})\|_{q \cdot l}.$

$$C(a) = \begin{cases} 1, & \text{if } a > C_2 \\ 0, & \text{if } a < C_1, 0 \leq C_1 \leq C_2 \\ \Delta, & \text{if } C_1 \leq a \leq C_2 \end{cases} \tag{21}$$

Since every pair $(S^u, j)$, such that $\tilde{S}^u \in K_j, S^u \in \tilde{S}^q$ is marked by some operator $B_v, 1 \leq v \leq w$, this operator is worth the evaluation $G_{uj} = G_j(S^u) \geq 1.$

All other systems either mark or do not mark this pair. Operators that do not mark a pair $(S^u, j)$ give an estimate that does not exceed an arbitrarily small value in absolute value $\delta$; therefore, one can choose $\delta$ such that

$$\delta < \frac{C_1}{W(C_1 + C_2)} \tag{22}$$

*W*—number of operators in the system $C_1, C_2$—parameters of the decision rule *C*.
Let us put

$$\tilde{B} = (C_1 + C_2)(B_1 + \ldots + B_w) \tag{23}$$

Consider what estimate the operator will build $\tilde{B}$ for an object $S^u$ from the class $K_j$. The couple $(S^u, j)$ is marked. This means that the operator $B_v$ constructs an estimate not less than 1. The remaining operators either also construct an estimate for this pair not less than 1, or an estimate less than $\delta$ and in turn, $\delta$ inequality Equation (22) holds. In the worst case, all other operators construct $\tilde{S}^q$ negative small estimates for to $K_j$. Then, if the estimate $S^u$ for $K_j$ in the operator $\tilde{B}$ is denoted by $\tilde{G}_{uj}$ then from Equation (23) it is easy to obtain the inequality:

$$\tilde{G}_{uj} \geq (C_1 + C_2) \cdot \left(1 - \frac{w-1}{w} \cdot \frac{C_1}{C_1 + C_2}\right) > (C_1 + C_2) \cdot \left(1 - \frac{C_1}{C_1 + C_2}\right) = C_2 \tag{24}$$

Applying the decision rule, we get that $C(\tilde{G}_{uj}) = 1$, and algorithm *A* establishes this inclusion.

This is true for any pair $(S^u, j)$ such that $S^u \in \tilde{S}^q, \tilde{S}^u \in K_j, j = 1, \ldots, l$ since all such pairs are marked by some operator from the system $B_1, \ldots, B_w$.

Now, let $S^u \in \tilde{S}^q$ be $\tilde{G}_{ur}$ the estimate built by the operator for the object $S^u$ according to the class $K_r$. Since $S^u$ does not belong, $K_r$ the pair $(S^u, r)$ is not marked in any of the operators $B_1, \ldots, B_w$. Consequently, each of these operators constructs an estimate for $S^u$ the class $K_r$ that does not exceed in absolute value $\delta$, from the definition of the operator $\tilde{B}$ and the estimate $\delta$ that $|\tilde{G}_{ur}| < (C_1 + C_2)W \cdot \left(\frac{C_1}{(C_1 + C_2)W}\right) = C_1.$

It can be seen from the definition of decision rule $C$ that the algorithm $\tilde{A} = \tilde{B} \cdot C$ establishes the inclusion $S^u \in K$, this is true for any pair $(S^u, r)$ such that $S^u \in \tilde{S}^q, S^u \in K_r, j = 1, 2, \ldots, l$.

**The theorem has been proven.** $\square$

The purpose of further constructions is to find a system of operators $B_1, \ldots, B_w$ that, for an arbitrary problem $Z = (T_{mnl}, \tilde{S}^q)$, mark any pair $(S^u, j)$ such that $S^u \in \tilde{S}^q, S^u \in K_j, j = 1, 2, \ldots, l$ and do not mark any pair $(S^u, K_r), S^u \in \tilde{S}^q, \tilde{S}^u \in K_r, j = 1, \ldots, l$.

If we manage to find such a system $B_1, \ldots, B_w$, then, according to $A$, the algorithm correctly solves problem $Z$. The correct algorithm for $Z$ can be written as:

$$\left( (C_1 + C_2) \sum_{i=1}^{w} B_i \right) \cdot C = A$$

Consider a sample $\tilde{S}^q = \{ S^i, \ldots, S^q \}, S^i = (b_{i1}, \ldots, b_{in})$ and a system of operators $B(j)$ such that all parameters $\varepsilon_{uv}, P_{uv}, u = 1, \ldots, m_{j-1}, m_{j+1}, \ldots, m, v = 1, 2, \ldots, n$ are chosen the same: $\varepsilon_{uv} = 0, P_{uv} = \varepsilon \geq 0$, where $\varepsilon$ is a sufficiently small number.

In other words, this means that if all pairs $\tilde{S}^u \in K_j(u, v)$ are assumed to be small, $\varepsilon_{uv}, P_{uv}, v = 1, 2, \ldots, m, u = m_{j-1} + 1, \ldots, m_j$ no restrictions are imposed on the parameters yet.

Let $B$ be an arbitrary operator from $B(j)$ and $S = (a_1, \ldots, a_n)$ an arbitrary admissible object.

Let also $B(S) = \left( G_1^j(S), \ldots, G_j^j(S), \ldots, G_l^j(S) \right)$.

**Lemma 1.** *Let* $0 \leq G_t^j(S) < m \cdot n \cdot \varepsilon, \quad t = 1, 2, \ldots, j - 1, j + 1, \ldots, l$.

**Proof of Lemma 1.** When forming a value, $G_t^j(S)$ only pairs $(u, v)$ are considered such that $u = m_{t-1} + 1, \ldots, m_t, \quad v = 1, \ldots, n$ only objects belonging to the class are considered $K_j$. $\square$

From these pairs, in turn, pairs are selected that are included in the reference set $\Omega_t$.

Let us denote the number of pairs from $K_t \bigcap \Omega_t$ through $n_t$. Obviously, there is an inequality: $n_t \leq n(m_t - m_{t-1}) < n \cdot m$

Since $P_{uv} = \varepsilon$ when comparing the elements $a$, (the value of the feature in the recognizable object $S$, $G_t^j(S)$ either 0 can be added to the value, if $p_v(a_{tv}, a_{uv}) > \varepsilon_{uv} = 0$, or the value $P_{uv} = \varepsilon$ if $p_v(a_{tv}, a_{uv}) \leq \varepsilon_{uv} = 0$.

If from the last assertion, as well as from the inequality, one easily obtains:

Consequence: Let $B^j \in B(j), B(Z) = B(T_{mnl}, S^q) = \left\| a_{rt}^j \right\|_{q \cdot l}$:

Then the elements—$a_{rt}^j$ at $t \neq j$ satisfy the inequalities:

$$0 \leq a_{rt}^j < n \cdot m \cdot \varepsilon, \quad r = 1, 2, \ldots, q$$

The proof of the corollary is obtained if we consistently apply Lemma 1 to recognizable objects $S^1, \ldots, S^q$ from the sample $\tilde{S}^q$.

We see that the operator $B$ from the family $(B)$ constructs numerical matrices in which the elements of all columns except the $j$-th can be made arbitrarily small with an appropriate choice of the value of $\varepsilon$. If it is required that $a_{rt}^j < \delta$, it is enough to put $\varepsilon = \frac{\delta}{n \cdot m}$.

Consider now a pair $(u, v)$ such that $1 \leq v \leq n$, in other words, an object $S_u \in \tilde{K}_j$. The pair $(u, v)$ corresponds to the element in the learning table $a_{uv}$. In the control sample, as was previously accepted, the objects belong to the class and the rest of the objects do not belong to the class. For brevity, we will assume that the objects that do not belong $\tilde{K}_j$ form the class $Q_j$, and the remaining objects form the class $CQ_j$. Consider the values $p_v(a_{uv}, b_{tv}), t = \overline{1, q}$, that is, the distance of the value $v$-th of the feature on the object $S_u$, in the learning table $T_{mnl}$ to the value of that feature in $S^t$ from $\tilde{S}^q$.

Let us arrange the objects from $\tilde{S}^q$ in ascending order of value $p_v(a_{uv}, b_{tv}), t = \overline{1,q}$, objects with equal values of elements, arrange among themselves in an arbitrary way.

We get:

$$S^{r_1}, S^{r_2}, \ldots, S^{r_i}, \ldots, S^{r_q}$$

$$0 \leq p_v(a_{uv}, b_{r_1 v}) \leq p_v(a_{uv}, b_{r_2 v}) \leq \ldots \leq p_v\left(a_{uv}, b_{r_q v}\right)$$

Let us assign to the objects of the sequence the sign "+" if they belong to the class and $Q_j$ the sign "−" if they belong to the class $CQ_j$. The result might be, for example, the following sequence:

$$S^{+r_1}, S^{+r_2}, S^{-r_3}, S^{-r_4}, S^{+r_5}, \ldots, \text{etc.}$$

**Definition 2.** *A pair $(u, v)$ is called stationary if in the constructed sequence the signs "+, −" take place equal to one change of sign, and if the change of sign occurs on the elements $S^{r_i}, S^{r_{i+1}}$ then:*

$$p_v(a_{uv}, b_{r_i v}) < p_v\left(a_{uv}, b_{r_{i+1} v}\right)$$

*Otherwise, the pair $(u, v)$ is called non-stationary.*

Let us first consider the case when for each number $j = 1, 2, \ldots, l$ there is at least one pair $(u, v)$ stationary. In this case, the basic operators from the family $B(j)$ are relatively easy to define, in different ways for the following two cases:

(A) The sequence has the form $S^{+r_1}, \ldots, S^{+r_i}, S^{-r_{i+1}}, \ldots, S^{-r_q}$, that is relatively $a_{uv}$ all objects of the class $Q_j$ are closer than all objects of class $CQ$, then the operator $B^j \in B(j)$ is redefined as follows:

    1.    The support set $\Omega_j$ is composed of one stationary pair $(u, v), 1 \leq v \leq n, u \in \{m_{j-1}, \ldots, m_j\}$.

    2.    $\varepsilon_{uv} = \frac{1}{2} p_v\left(a_{uv}, b_{r_{i+1}}, V\right)$ other $\varepsilon_{rw} = 0$.

    3.    $P = N$ otherwise $P_{uw} = \varepsilon'_j$, $\varepsilon'_j$—a fairly small value.

(B) The sequence looks like $S^{-r_t}, \ldots, S^{+r_i}, S^{+r_{i+1}}, \ldots, S^{+r_q}$. The operator is searched in the form $B^{j_1} - B^{j_2}; B^{j_1}, B^{j_2} \in B(j)$. As before, in both operators, the support set $\Omega$, is composed of one critical pair $(u, v)$. $P_{uv} = N$, other $P_{rw} = \varepsilon'$. But:

In operator $B^{j_1} : \varepsilon_{uv} = p_v\left(a_{uv}, b_{r_q v}\right) + 1$.

In operator $B : \varepsilon_{uv} = \frac{1}{2}\left[p_u\left(a_{uv}, b_{r_{i-1} v}\right) + p_v\left(a_{uv}, b_{r_q}\right)\right]$.

Thus, we completely defined the operators $B_1 \in B(1), B_1 \in B(l)$.

Let the result of applying the operator $B^j$ to the problem $Z = (T_{mnl}, S^q)$ be denoted by $\|b_{rt}\|_{q \cdot l}$.

**Lemma 2.** *If $S^r \in CQ\left(S^r \in \tilde{S}^q\right) \bigcap K_j$ then if $S^r \in CQ$ then $b^j_{rj} = 0$.*

**Proof of Lemma 2.** 1. Consider the first case of defining the operator $B^j$. Then, there is a stationary pair $(u, v)$ for which the descriptions of the above sequence are as follows:

$$S^{+r_1}, \ldots, S^{+r_w}, S^{-r_{w+1}}, \ldots, S^{-r_q}$$

where

$$Q = \{S^{r_1}, \ldots, S^{r_w}\}, \quad CQ = \{S^{r_{w+1}}, \ldots, S^{r_q}\}, \quad p_v(a_{uv}, b_{r_w v}) < p_v\left(a_{uv}, b_{r_{w+1} v}\right) \quad (25)$$

When defining the operator $B$, the quantities $\varepsilon_v$ are chosen in such a way that the following inequalities are satisfied:

$$p_v(a_{uv}, b_{r_w v}) < \varepsilon_v < p_v\left(a_{uv}, b_{r_{w+1} v}\right) \quad (26)$$

From inequality [19], it is easy to see that for each object from $Q$ the proximity function for this $S$ over the reference set $S^i$, $\Omega = \{(u, v)\}$ is equal to 1. Therefore, $b_{ij} = N$, $i = r_i, \ldots, r_w$.

Similarly, the proximity function for objects from $CQ$ is $\Omega = \{(u, v)\}$ equal to 0. And therefore, $b_{ij}^j = 0$, $i = r_w + 1, \ldots, r_q$.

2. Consider the second case in the definition of the operator $B^j$. In this case, there is a stationary point $(u, v)$ and the sequence corresponding to it has the form:

$$S^{-r}, \ldots, S^{-rw}, S^{+rw+1}, \ldots, S^{+rq}; \quad Q = \left\{ S^{rw+1}, S^{rq} \right\}, CQ = \left\{ S^{r1}, \ldots, S^{rw} \right\} \quad (27)$$

In the operator, $B^j$ the value $\varepsilon_v$ is chosen in such a way that the proximity function for the reference value $\Omega_j = \{(u, v)\}$ is equal to 1 for all objects from the selection $\tilde{S}^q$.

Therefore, if $B^{ij}(Z) = \left\| b_{rt}^{ij} \right\|_{q \cdot l}$, then $b_{rt}^{ij} = N$, $r = \overline{1, q}$.

In the operator $B^{jr}(Z) = \left\| b_{rt}^{ji} \right\|_{q \cdot l}$, the value $\varepsilon_v$ is chosen in such a way that the proximity function in the reference set $\Omega_j = \{(u, v)\}$ is equal to 1 for objects from $CQ$, and is equal to 0 for objects from $Q$. That is why

$$\begin{cases} b_{rj}^{jr} = N, & S^r \in CQ \\ b_{rj}^{jr} = 0, & S^r \in Q \end{cases} \quad (28)$$

From equality [21,22], and also from the fact that $B^j = B^{j1} - B^{j2}$ equalities follow:

$$\begin{cases} b_{rj}^{j} = N, & S^r \in CQ \\ b_{rj}^{j} = 0, & S^r \in Q \end{cases} \quad (29)$$

**The Lemma is proven.** □

Consider now the operator: $B = B^1 + \ldots + B^l$ and put $B(Z) = B\left(T_{mnl}, \tilde{S}^q\right) = \|b_{rt}\|_{q \cdot l}$.

In Lemma 2, when applying the operator, $B^j$, the elements of all columns with the exception of $j$ are not exceeded.

Objects $S^r \in K_j$ obtain grade $N$.

Objects $S^r \in K_j \cap \tilde{S}^q$ obtain grade 0.

Because $b_{rt} = b_{rt}^1 + \ldots + b_{rt}^l$ by definition of the operator $B^j$, then if

$$S^r \in K_j \cap \tilde{S}^q \Rightarrow b_{rj} \geq N + m \cdot n \cdot (l - 1) \cdot \varepsilon \quad (30)$$

$$S \in CK_j \cap \tilde{S}^q \Rightarrow |b_{rj}| \leq m \cdot n \cdot l \cdot \varepsilon \quad (31)$$

Having appropriately chosen the values, $N, \varepsilon$ and using Theorem 3, we prove the theorem.

**Theorem 3.** *If in the problem $Z$ for each number $j = \overline{1, l}$ there is a stationary pair, then the algorithm $A = \left( (C_1 + C_2) \sum\limits_{i=1}^{l} B^i \right) \cdot (C(C_1, C_2))$ is correct for problem $Z$.*

Since the operator we have constructed $(C_1 + C_2) \sum\limits_{i=1}^{l} B^i$ belongs to the linear closure of a model of the type of calculation of estimates, then we proved the completeness of the linear closure of this model for all problems in which for each class there is at least one stationary pair, $(u, v)$, and moreover, we wrote this correct algorithm $A$ explicitly.

The verification of the fact that such a stationary pair really exists is not difficult.

For this, enough for each pair $(u, v)$, where $u = m_{j-1} + 1, \ldots, m_j$, $v = 1, 2, \ldots, n$ calculate all $p_v(a_{uv}, b_{iv})$, $i = 1, 2, \ldots, q$ and check whether all inequalities of the 1st or 2nd group are fulfilled simultaneously. If all inequalities of groups 1 and 2 are simultaneously

satisfied, then the pair $(u, v)$ is stationary and can be constructed $B^j$ in the same way as was performed in the proof of the theorem.

The conditions for the existence of a stationary pair essentially mean the following:

There is an element $S^u$, $S^u \in K_j$ and features in the learning table $T_{mnl}$.

1. The distance according to the $V$ attribute from $S_v$ to all elements of the control sample that belong is $K_j$ strictly less than all such distances for objects of the control sample that do not belong to $K$;

2. The distance according to the $V$ attribute from the object in the control sample that does not belong is $K_j$ less than all such distances for the objects of the control sample that belong to $K_j$.

Consider the 2nd case, when the construction of a correct algorithm in a linear closure is quite simple. As before, we arrange the objects of the control set $\tilde{S}^q$ in sequence by increasing the distance from some value of the feature u, in the reference $S_w$, $S_w \in \tilde{K}_j$, $1 \leq u \leq n$. We put the constructed sequence over the elements of the sequence with the sign "+", if, $S^{ri} \in K_j$ and the sign "−", if $S^{ri} \in CK_j$ denoted by $\pi(w, u)$. Let us find in this sequence the last element in order $S^{+r_p}$, this element must also be such that the next element $S^{-r_p}$ has the property:

$$p_v(a_{wu}, b_{r_t u}) < p_v\left(a_{wu}, b_{r_p u}\right)$$

The set of elements of the sequence $\pi(w, u)$ following the element $S^{+r_t}$ is denoted by $M^-(w, u)$.

We introduce the set $M_j^- = \bigcup\limits_{S_w \in \tilde{K}_j} \bigcup\limits_{u=1}^{n} M^-(w, u)$.

Similarly to the previous one, in each sequence, we select the first element in order $S^{+r_t}$, and moreover, such that $S^{-r_p}$ the inequality holds for the previous element:

$$p_v\left(a_{wu}, b_{r_p u}\right) < p_v(a_{wu}, b_{r_t u})$$

Element $S^{+r_t}$ in the sequence $\pi(w, u)$ is denoted by $M^+(w, u)$. We introduce the set

$$M_j^+ = \bigcup\limits_{S_w \in \tilde{K}_j} \bigcup\limits_{u=1}^{n} M^+(w, u) \tag{32}$$

**Definition 3.** *The problem $Z = \left(J, \tilde{S}^q\right)$ is called monotonic if for each $j = 1, 2, \ldots, l$ one of the two equalities is satisfied:*

$$\tilde{S}^q \bigcap C\tilde{K}_j = M_j^- \tag{33}$$

$$\tilde{S}^q \bigcap \tilde{K}_j = M_j^+ \tag{34}$$

The meaning of the monotonicity condition is as follows, an arbitrary reference object is chosen to belong to $\tilde{K}_j$ and an arbitrary feature with number $u$, in the control sample $\tilde{S}^q$ we select all objects that do not belong to the class $\tilde{K}_j$ and such that, relative to the selected feature, they are farther from $S_w$ than all objects $\tilde{S}^q$ belonging to the class $\tilde{K}_j$.

The set of all such objects not belonging to the class $\tilde{K}_j$ is denoted by $M^-(w, u)$. Condition [17] is that if all elements of the set $M_j^-(w, u)$ are summed up, then all elements from the control sample that do not belong to the class are obtained $\tilde{K}_j$.

Construction of operators $B_j, j = 1, 2, \ldots$, for a monotonic problem $Z$.

1. The choice of the reference subset $\Omega$, since the problem $Z$ is monotone, then one of the conditions [17] or [18] is satisfied, the choice of the reference set is the same in both cases, so we will assume that we have:

$$\tilde{S}^q \bigcap CK_j = M_j^- \bigcup\limits_{S_w \in \tilde{K}_j} \bigcup\limits_{u=1}^{n} M^-(w, u) \tag{35}$$

The set $M^-(w, u)$ forms a cover of the set $\tilde{S}^q \bigcap CK$, but this cover may be redundant, in other words, some sets $M^-(w, u)$ may be removed so that the cover remains a cover. Removing such extra sets, we construct irreducible covers for $\tilde{S}^q \bigcap CK_j$.

Let the constructed irreducible cover have the form:

$$\tilde{S}^q \bigcap CK_j = M^-(w_1, u_1) \bigcup \ldots \bigcup M^-(w_2, u_2) \tag{36}$$

Then, $\Omega_j = \{(w_1, u_1), \ldots, (w_2, u_2)\}$.

Choice of options $\tilde{\varepsilon}_j$.

All parameters of this group, with the exception of the parameters, $\varepsilon_{w_i u_i}$, $i = 1, 2, \ldots$, are chosen so large that the following inequalities are satisfied: $p_v(a_{rv} b_{iv}) < \varepsilon_{rv}$, $i = 1, 2, \ldots, q$ or $\varepsilon_{rv} = \max\limits_{i = \overline{1, q}} p_v(a_{rv}, b_{iv}) + 1$.

The parameters $\varepsilon_{w_i u_i} = \frac{1}{2}(p_{ui}(a_{w_i u_i} b_{tu_i}) + p(a_{w_i u_i} b_{tu_i}))$ are chosen so that the following inequalities hold:

In the sequence $\pi(w_i, u_i)$, we find the last element $S^{+p}$ and the element following it $S^{-t}$, then $p_{u_i}(a_{w_i u_i} b_{pu_i}) < \varepsilon_{w_i u_i} < p_{u_i}(a_{w_i u_i} b_{tu_i})$ or $\varepsilon_{w_i u_i} = \frac{1}{2}(p_{u_i}(a_{w_i u_i} b_{tu_i}) + p(a_{w_i u_i} b_{tu_i}))$.

This choice of parameters $\varepsilon_{w_i u_i}$ occurs in cases where the relation is satisfied: $\tilde{\varepsilon}^q \bigcap CK_j = M_j^-$. If this relation is not satisfied, but the relation $\tilde{\varepsilon}^q \bigcap K_j = M_j^+$.

Then, the operator $B_j$ is sought in the form of a difference $B_{j_1} - B_{j_2}$, and the values $\varepsilon_{ij}$ are chosen differently for the operators that make up the difference.

Options $\tilde{p}\tilde{\gamma}$. The parameters $\gamma(S_i) = \gamma_i$ are assumed to be equal to a sufficiently small value; $\delta$ more precisely, the value $\delta$ will be indicated later for all $S_i \in \tilde{K}_j$ and also for such $S_i$ numbers of which are not contained in the set $W_1, \ldots, W_2$. For the rest $S_i$, we assume: $\gamma(S_i) = \gamma_i = N$.

Similarly, $P_u = \ldots = P_{ur} = N$ rest $P_i$, $P_i = \delta$. The values $N$ are chosen to be sufficiently large; the exact values for $N$ are given later. The operator definition $B_j$ is complete for the case: $\tilde{S}^q \bigcap CK_j = M_j^-$.

**Theorem 4.** *The operator for calculating estimates defined above marks all objects from the class in the control sample $K_j$ and only them. In other words, if $B(Z) = \left\| G_{ij} \right\|_{q \cdot l}$ and objects $S^{u_1}, \ldots, S^{u_n} \in K_j$ at the remaining objects do not belong to the control sample $K_j$, then $G_{u_t j} \geq N$, $t = \overline{1, K}$ $\left| G_{vj} \right| < \delta$, $V \notin \{u_1, \ldots, u_k\}$.*

**Proof of Theorem 4.** Threshold parameters $\varepsilon_{uv}$ and reference sets were chosen in such a way that for each object $S^{up} = (b_{up_1}, \ldots, b_{up_n})$ from $K_j$ the relation was fulfilled $p_v\left(a_{uv} b_{u_p v}\right) \leq \varepsilon_{uv}$ in such a way, according to the constructed support set, the proximity function is equal to 1, and the total estimate is not less than $\sum p_{uv} \cdot \gamma_{uv} \geq N \cdot N = N^2 > N$.

If the object of the control set does not belong, then $K_j$ it has a mark $(-)(u, v)$ in the sequences, and therefore there is such a pair $\pi$(by the definition of a monotone problem) that in the sequence $\pi(u, v)$ this element is after all elements $K_j$, then $\varepsilon_{uv}$ they are chosen so that the corresponding inequality $p_v(a_{uv} b_{iv}) \leq \varepsilon_{uv}$ violated. In this case, the proximity function for the object $S^i \in C\tilde{K}_j$ is equal to 0, and $G_{ij} = 0 < \delta$.

The evaluation operator constructed in this way puts large estimates for control elements from $\tilde{K}_j$, and small estimates for elements from $C\tilde{K}_j$. The last assertion easily implies the correctness of the algorithm composed of the previously defined operator and the threshold decision rule.

**The Theorem has been proven.** $\square$

The proof for the 2nd case of the monotone problem is carried out according to the same principle (see the proof for the stationary problem) only the operator is sought as the difference between two operators.

The simple cases presented by us are an illustration for the proof of the main theorem on the well-posedness of the linear closure. The proof of this theorem is technically more

complicated, it is divided into a series of steps; however, each individual step implements a construction of the same type that was used in the last theorems.

When proving the main theorem, in essence, it is the linear closure that is used.

The proof of the linear closure well-posedness theorem will be carried out in two stages. In the first stage, we will introduce one additional constraint on the control objects of the training sample.

Practical tasks really satisfy him. In the first stage, the well-posedness theorem will be proved under this constraint.

**Definition 4.** *Control objects* $S^1, \ldots, S^q$ *are called consistent with objects* $S_1, \ldots, S_m$ *if for each pair* $S^u, S^v, S^u \in K_j, S^v \notin K_j$ *there is at least one object* $S_r \in K_j$ *and attribute* $W$ *such that* $p_w(a_{rw}, b_{uw}) < p_w(a_{rw}, b_{vw})$.

The lack of consistency in the control sample means that in the control there are objects $S^v \notin K_j$ that are closer in all respects to all objects in the class $K_j$. In this case, all learning information is collected in such a way as to assign a closer object $S^v$ to a class $K_j$ with greater preference than a more distant object $S^q$ and thereby make a mistake.

We will first consider classes in which there are no such pathological cases, that is, the situation when the control objects are consistent with the training ones.

Let, as before, the objects $S_1, \ldots, S_m; S_i = (a_{i1}, \ldots, a_{in})$ $i = \overline{1, n}$.

They form learning information $J$, and the objects $S_{m_{j-1}} + 1, \ldots, S_{m_j}$, $m_0 = 0, m_l = m$ belong to the class $K_j$ and do not belong to other classes.

The objects $S^1, \ldots, S^q, S^i = (b_{i1}, \ldots, b_{in})$, $j = \overline{1, l}$, $i = \overline{1, q}$ form a control sample and the objects $S^{q_{j-1}+1}, \ldots, S^{q_i}$ belong to a class $K_j$ and do not belong to other classes.

As before, we denote:

$$\{S_1, \ldots, S_m\} \bigcap K_j = K, \quad \{S_1, \ldots, S_m\} \backslash \tilde{K}_j = C\tilde{K}_j \tag{37}$$

$$\left\{S^1, \ldots, S^q\right\} \bigcap K_j = K_j^1, \quad \left\{S^1, \ldots, S^q\right\} \backslash \tilde{K}_j^1 = CK_j^1 \tag{38}$$

Operator formation $B_j, j = \overline{1, l}$.

1. For all pairs $(u, v)$ such that $S_k \in C\tilde{K}_j$, $V = \overline{1, K}$ we assume $\varepsilon_{uv} = 0$, $p_{uv} = \gamma_{uv} = \delta$.
2. Consider an arbitrary control object $S^t \in K_j^1$.

By the definition of a training-consistent control sample, for each $S^r \in K_j^1$ there is such a pair $(S_k, W), S_k \in K_j, 1 \leq W \leq n$.

What $p_w(a_{kw}, b_{tw}) < p_w(a_{kw}, b_{rw})$. For the pair $(k, w)$ we assume:

$$p_{kw} = N, \quad \gamma_{kw} = N, \quad \varepsilon_{kw} = \tfrac{1}{2}(p_w(a_{kw}, b_{tw}) + p_w(a_{kw}, b_{rw})) \tag{39}$$

Obviously $p_w(a_{kw}, b_{tw}) < \varepsilon_{kw} < p_w(a_{kw}, b_{rw})$.

Let us include in the support set all pairs $(k, w)$ for all $S^r \in K_j^1$.

We have defined an auxiliary operator: $B_j^t, 1 \leq j \leq l, S^t \in K_j^1$

$$B_j = \sum_{S^t \in K_j^1} B_j^t, \quad j = 1, 2, \ldots, l \tag{40}$$

Formation of operator $B$. $B = \sum\limits_{j=1}^{l} B_j$

We need to study what numeric matrix the operator $B$ translates into task $Z$, with a training sample $S_1, \ldots, S_m$ and a control sample $S^1, \ldots, S^q$. The evaluation of the elements of this matrix will be performed sequentially by the steps of constructing the operator $B$. The first step is to form the operator $B_j^t$.

Let $B_j^t(Z) = \left\| G_{\alpha\beta}(j, t) \right\|_{q \cdot l}$.

**Lemma 3.** *If $\beta \neq j$ that $G_{\alpha\beta}(j,t) \cdot \delta^2 \cdot n \cdot m$.*

**Proof of Lemma 3.** The formation of estimates with respect to the class $B_\beta$ is carried out for some of the pairs $(u,v)$ such that for each such pair the term included in the estimate does not exceed: $P_u \gamma_n = \delta \cdot \delta = \delta^2$.

The total number of pairs $(u,v)$, $S_u \in CK_j$ is obviously $n \cdot (m - (m_j - m_{j-1})) < n \cdot m_j$.

From this, it easily follows that $G_{\alpha\beta}(j,t) \cdot \delta^2 \cdot n \cdot m$.

**The Lemma is proven.** □

**Lemma 4.** *If $\beta \neq j, S^\alpha \in CK_j$, that $G_{\alpha\beta}(j,t) = 0$.*

**Proof of Lemma 4.** In the base set of the operator $B_j^t$ for each element $S^\alpha \in CK_j$ there is a pair $(k,w), S_k \in K_j, 1 \leq w \leq n$ such that $p_w(a_{kw}, b_{tw}) < p_w(a_{kw}, b_{\alpha w})$.

By choosing the parameters $E_{kw}$, as shown above, we obtain $p_w(a_{kw}, b_{tw}) > \varepsilon_{kw}$.

According to the definition of the proximity function for the unique support set of the operator $B_j^t$, for the object $S^\alpha$ it is equal to 0. Therefore, $G_{\alpha\beta}(j,t) \geq 0$, which proves the Lemma. □

**Lemma 5.** *$G_{\alpha\beta}(j,t) \geq N^2$.*

**Proof of Lemma 5.** For each pair $(k,w)$ included in the base set of the operator, $B_j^t$ the parameter $\varepsilon_{kw}$, is chosen so that $p(a_{kw}, b_{tw}) > \varepsilon_{kw}$. Due to inequality, the proximity function in the base set of the operator $B_j^t$ is equal to 1; therefore, $G_{tj}(j,t) = \sum\limits_{(k,w)} P_{kw} \gamma_{kw} = \sum N \cdot N > N^2$.

**The Lemma is proven.** □

Let $B_j(Z) = \left\| G_{\alpha\beta}(j) \right\|_{q \cdot l}, \ j = \overline{1,l}$.

By definition of the operator $B_j$.

$$G_{\alpha\beta}(j) = G_{\alpha\beta}(j, q_{j-1} + 1) + \ldots + G_{\alpha\beta}(j, q_j) \tag{41}$$

Using the matrix $\left\| G_{\alpha\beta}(j,t) \right\|_{q \cdot l}$ and Equation (41), we obtain the following inequalities:

$$\begin{cases} G_{j\alpha}(j) \geq N^2, \alpha = q_{i-1} + 1, \ldots, q_i \\ G_{j\alpha}(j) = 0, \alpha = 1, 2, \ldots, q_{i-1}, q_{i+1}, \ldots, q \end{cases} \tag{42}$$

True $0 \leq G_{\alpha\beta}(j) \leq q \cdot n \cdot \delta^2$ for others $\alpha\beta$.

$$\begin{array}{c} S \\ S' \\ \vdots \\ S^{q_{j-1}} \\ S^{q_{j-1}+1} \\ \vdots \\ S^{t-1} \\ S^t \\ S^{t+1} \\ S^{q_j} \\ \vdots \\ S^q \end{array} \left\| \begin{array}{cccccccc} 1 & 2 & \ldots & j-1 & j & j+1 & \ldots & l \\ 0 & 0 & 0 & 0 & 0 \leq G_{\alpha\beta}(j) \leq q \cdot n \cdot \delta^2 & 0 & 0 & 0 \\ \ldots & \ldots & \ldots & \ldots & \ldots & \ldots & \ldots & \ldots \\ 0 & 0 & 0 & 0 & 0 & 0 & 0 & 0 \\ 0 & 0 & 0 & 0 & \geq N^2 & 0 & 0 & 0 \\ \ldots & \ldots & \ldots & \ldots & \ldots & \ldots & \ldots & \ldots \\ 0 & 0 & 0 & 0 & \geq N^2 & 0 & 0 & 0 \\ \ldots & \ldots & \ldots & \ldots & \ldots & \ldots & \ldots & \ldots \\ 0 & 0 & 0 & 0 & \geq N^2 & 0 & 0 & 0 \\ 0 & 0 & 0 & 0 & 0 & 0 & 0 & 0 \\ \ldots & \ldots & \ldots & \ldots & \ldots & \ldots & \ldots & \ldots \\ 0 & 0 & 0 & 0 & 0 & 0 & 0 & 0 \end{array} \right\|$$

Because $B = \sum\limits_{j=1}^{l} B_j$.

And $\left\|G_{\alpha\beta}\right\|_{q\cdot l}$ is the matrix into which the operator transforms the problem, then $G_{\alpha\beta} = \sum\limits_{j=1}^{l} G_{\alpha\beta}(j), \ \alpha = \overline{1,q}, \ \beta = \overline{1,l}.$

Lemmas 3–5 clearly imply the following inequalities $G_{\alpha\beta} \geq 1 \cdot N^2, \beta = \overline{1,l}$ at $\beta = j, \alpha = q_{j-1} + 1, \ldots, q; \ q_0 = q, q_1 = q.$

For everyone else $\alpha, \beta: 0 \leq G_{\alpha\beta} \leq n \cdot m \cdot l \cdot \delta^2.$

$C_1, C_2$ is a decision rule; then, we can choose the values in such a way $\delta, N$ that the algorithm $A = B \cdot C(C_1, C_2)$ will give correct answers for all elements $S^1, \ldots, S^q$ across all classes $K_1, \ldots, K_l.$

Recall that the decision rule $(C_1, C_2)$ is applied to numerical matrices element by element, that is, $C\left(\left\|G_{\alpha\beta}\right\|_{q\cdot l}\right) = \left\|C\left(G_{\alpha\beta}\right)_{q\cdot l}\right\|$ and $C(G_{\alpha\beta}) = \begin{cases} 1, G_{\alpha\beta} > C_2 \\ 0, G_{\alpha\beta} < C_1, \ 0 < C_1 < C_2 \ . \\ \Delta, C_1 \leq G_{\alpha\beta} \leq C_2 \end{cases}$

From the definition of the decision rule and inequalities, it is clear that the parameter $N$ must be chosen so that the inequality $1 \cdot N^2 > C_2.$

Therefore, one can put $N = \sqrt{\frac{C_2}{l}} + 1.$

With this choice of the parameter $N$, each of the objects $S^{q_{j-1}+1}, \ldots, S^{q_j}$ will be assigned by the algorithm $A$ to the class $K_j.$

It follows from the definition of the threshold decision rule and inequalities that the parameter $\delta$ can be chosen so that the inequality $n \cdot m \cdot l \cdot \delta^2 < C_1.$

Therefore, it suffices to take as $\delta$ any positive quantity satisfying the inequalities $\delta < \sqrt{\frac{C_1}{n\cdot m\cdot q\cdot l}}.$

In this case, each of the objects $S^1, \ldots, S^{q_{j-1}} S^{q_{j+1}}, \ldots, S^q$ will not be assigned to the class $K_j, j = \overline{1,l}, q_1 = 0, q_l = q.$

We have proven.

**Theorem 5.** $A = B \cdot C(C_1, C_2)$ *with threshold decision rule C and operator B.*

$$B = \sum_{j=1}^{l} B_j, \ \ B_j = \sum_{S^t \in K_j} B_j^t$$

Operators $B_j^t$ are operators for calculating estimates, correct for the problem $Z = \left\{S_1, \ldots, S_m, \left\|\alpha_{ij}\right\|_{m\cdot l}, S^1, \ldots, S^q\right\}.$

Each of the operators $B_j$ is specified by a set of numerical parameters $\varepsilon_{uv}, p_{uv}$, where $u = \overline{1,m}, v = \overline{1,n}.$ $m$—the number of objects in the learning sample, $n$—the number of features participating in the description of objects, that is, a set of $3 \cdot n \cdot m$ numerical parameters.

From all that has been said above, it follows that with the described choice of parameters $N$, the matrix $\left\|\beta_{ij}\right\|_{q\cdot l}$, into which the problem $Z$ is converted by the algorithm $A = B \cdot C(C_1, C_2).$

As follows:

$$
\begin{array}{c}
S^1 \\
S^{q_1} \\
\vdots \\
S^{q_1+1} \\
S^{q_j} \\
\\
\vdots \\
S^{q_j+1} \\
\\
\vdots \\
S^{q_{l-1}} \\
\\
\vdots \\
S^{q_{l-1}+1} \\
\\
\vdots \\
S^q
\end{array}
\left\| \begin{array}{cccccc}
1 & 2 & \ldots & j & \ldots & l \\
1 & 0 & \ldots & 0 & \ldots & 0 \\
\ldots & \ldots & \ldots & \ldots & \ldots & \ldots \\
1 & 0 & \ldots & 0 & \ldots & 0 \\
0 & 1 & \ldots & 0 & \ldots & 0 \\
\ldots & \ldots & \ldots & \ldots & \ldots & \ldots \\
0 & 0 & \ldots & 1 & \ldots & 0 \\
\ldots & \ldots & \ldots & \ldots & \ldots & \ldots \\
0 & 0 & \ldots & 0 & \ldots & 0 \\
\ldots & \ldots & \ldots & \ldots & \ldots & \ldots \\
0 & 0 & \ldots & 0 & \ldots & 1 \\
\ldots & \ldots & \ldots & \ldots & \ldots & \ldots \\
0 & 0 & \ldots & 0 & \ldots & 1
\end{array} \right\|
$$

Here, the operator $B$ in terms of elementary operators has the form $B = \sum_{j=1}^{l} \sum_{S^t \in K_j} B_j^t$ that is, the operator $B$ belongs to the linear closure of the previously introduced families of estimation algorithms.

Significant memory is required to write code $B$; however, when solving real applied problems, the operator code can be placed in the RAM of modern computers. So, if $n = 100, m = 50, q = 100$, that is, the training material consists of 50 objects described by 100 features, and it is required to store $1.5 \times 10^{76}$ numbers in memory.

In the future, we will consider methods that make it possible to more economically encode the operator B, which will reduce the required memory and more efficiently use the constructed correct algorithms for solving applied problems.

Summarizing all the above, we can state that we have proved the theorem.

**Theorem 6.** *Algorithm*

$$
A = \left( \sum_{j=1}^{l} \sum_{S^t \in K_j} B_j(S^t) \cdot C(C_1, C_2) \right)
$$

*with an operator from the linear closure of the score, the calculation model is correct for problem Z, and the operator is the sum of q operators from the score calculation model and is described by a set of $3 \cdot n \cdot m \cdot q$ numerical parameters.*

The only condition for constructing a correct rule is the consistency of the training control sample.

## 7. Results

In [15], the use of Internet of Things technologies in ecology (using the example of the Aral Sea region) as a system is proposed (Figure 1). The system was developed as a website. The Django framework was used to develop the server side of the system and Vue.js was used to develop the client side as a single page application (SPA) using the Quasar GUI framework. To start the system, you need to open this website ecoaral.uz in one of the modern browsers. After this, the main window of the system will be reflected on the browser page.

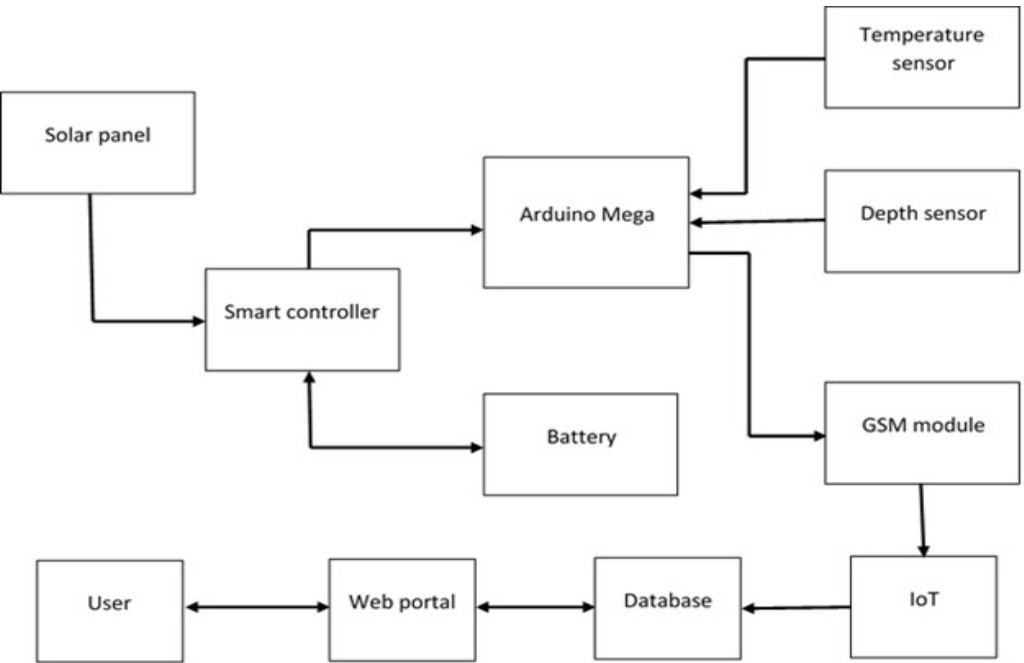

**Figure 1.** Scheme of functioning of the IoT environmental monitoring system.

In this article, to identify input information flows, we considered the so-called parametric recognition algorithms, i.e., such sets of algorithms in which each algorithm is one-to-one encoded by a set of numerical parameters. These models analyze the proximity between parts of previously classified objects and the object to be classified. Based on a set of assessments, a general assessment of the object is generated and, according to the introduced decision rule, the belonging of the recognized object to one or another class is determined.

In this article, as the initial model ($A$), we consider a model related to the model for calculating estimates, supplemented with some simple recognition algorithms such as the nearest neighbor algorithm, the average distance algorithm, etc.

The peculiarity of algorithms of this class is that in order to calculate estimates that determine the identity of a recognized object, there are simple analytical formulas that replace complex enumeration procedures that arise when calculating proximity estimates using a system of support sets.

In these models, the division of the algorithm into recognition operators and decision rules is carried out in a natural way.

We will only consider algorithms that can be represented in the form $A = B \cdot C$, where $B$ is an arbitrary recognition operator. It turns out that an essential part of the algorithm is the operator—$B$; the decisive rule is that $C$ can be made standard for all algorithms and programs. Any voting recognition operator maps task $Z$ into a numeric matrix of votes or ratings $B(Z) = \left\| \Gamma_{ij} \right\|_{q \cdot l} \Gamma_{ij} = \Gamma_j(S)$

Moreover, the value $\Gamma_{ij}$ has a clear, meaningful interpretation. This value can be considered as the degree of belonging of the examined object $S^i$ to class $K_j$, expressed as a number. Let

$$(\tilde{\alpha} - \tilde{\beta}) = \sum_{i=1}^{6} \sum_{j=1}^{10} |\alpha_i - \beta_j| \to \min \tag{43}$$

$\alpha_i$—these are sensor data, $\beta_j$—salinity class parameters. Sensor data in Table 2: $HCO_3$ (bicarbonate), $Cl$ (chlorine), $SO_4$ (sulfuric acid), $Ca$ (calcium), $Mg$ (magnesium), $Na$ (sodium) in Tables 3 and 4: Salinity is divided into five classes (non-saline, slightly saline, moderately saline, highly saline, and very highly saline), and each class consists of 10 points.

If the sensor receives data $a_i$ and it checks the value $a_i - b_i \to \min$ if the given condition completed, and indicates which salinity class it belongs to. If the $a_i - b_i \to \min$ values

correspond to more than one salinity class, then the information was intercepted somewhere (see Figure 2).

**Table 2.** Number of parameter votes.

| Valid $(HCO_3)$ | Valid $(Cl)$ | Valid $(SO_4)$ | Valid $(Na)$ | Frequency | Percent | Valid Percent | Cumulative Percent |
|---|---|---|---|---|---|---|---|
| 0.68 | 0.56 | 0.50 | 0.51 | 1 | 4.0 | 4.0 | 4.0 |
| 0.73 | 0.67 | 0.60 | 0.58 | 1 | 4.0 | 4.0 | 8.0 |
| 0.78 | 0.67 | 0.68 | 0.68 | 1 | 4.0 | 4.0 | 12.0 |
| 0.89 | 0.68 | 0.68 | 0.78 | 1 | 4.0 | 4.0 | 16.0 |
| 0.93 | 0.68 | 0.69 | 0.80 | 1 | 4.0 | 4.0 | 20.0 |
| 0.94 | 0.68 | 0.69 | 0.80 | 1 | 4.0 | 4.0 | 24.0 |
| 0.95 | 0.68 | 0.69 | 0.80 | 1 | 4.0 | 4.0 | 28.0 |
| 0.95 | 0.68 | 0.69 | 0.80 | 1 | 4.0 | 4.0 | 32.0 |
| 0.95 | 0.70 | 0.69 | 0.81 | 1 | 4.0 | 4.0 | 36.0 |
| 0.95 | 0.70 | 0.70 | 0.81 | 1 | 4.0 | 4.0 | 40.0 |
| 0.95 | 0.71 | 0.71 | 0.81 | 1 | 4.0 | 4.0 | 44.0 |
| 0.96 | 0.71 | 0.72 | 0.81 | 1 | 4.0 | 4.0 | 48.0 |
| 0.96 | 0.74 | 0.72 | 0.81 | 1 | 4.0 | 4.0 | 52.0 |
| 0.96 | 0.74 | 0.74 | 0.81 | 1 | 4.0 | 4.0 | 56.0 |
| 0.96 | 0.76 | 0.75 | 0.82 | 1 | 4.0 | 4.0 | 60.0 |
| 0.96 | 0.78 | 0.77 | 0.82 | 1 | 4.0 | 4.0 | 64.0 |
| 0.97 | 0.82 | 0.81 | 0.82 | 1 | 4.0 | 4.0 | 68.0 |
| 0.97 | 0.83 | 0.82 | 0.82 | 1 | 4.0 | 4.0 | 72.0 |
| 0.97 | 0.85 | 0.83 | 0.83 | 1 | 4.0 | 4.0 | 76.0 |
| 0.98 | 0.88 | 0.84 | 0.84 | 1 | 4.0 | 4.0 | 80.0 |
| 0.98 | 0.89 | 0.86 | 0.85 | 1 | 4.0 | 4.0 | 84.0 |
| 0.98 | 0.91 | 0.87 | 0.87 | 1 | 4.0 | 4.0 | 88.0 |
| 0.98 | 0.92 | 0.89 | 0.87 | 1 | 4.0 | 4.0 | 92.0 |
| 0.99 | 0.96 | 0.90 | 0.93 | 1 | 4.0 | 4.0 | 96.0 |
| 1.00 | 1.00 | 0.98 | 0.96 | 1 | 4.0 | 4.0 | 100.0 |
| **Total** | | | | **25** | **100** | **100** | |

**Table 3.** Descriptive Statistics.

|  | No. | Range | Minimum | Maximum | Mean | Std. Deviation | Variance |
|---|---|---|---|---|---|---|---|
| $\rho_i(HCO_3)$ | 25 | 0.32 | 0.68 | 1.00 | 0.9323 | 0.08130 | 0.007 |
| $\rho_i(Cl)$ | 25 | 0.44 | 0.56 | 1. 00 | 0.7675 | 0.11096 | 0.012 |
| $\rho_i(SO_4)$ | 25 | 0.48 | 0.50 | 0.98 | 0.7528 | 0.10544 | 0.011 |
| $\rho_i(Na)$ | 25 | 0.45 | 0.51 | 0.96 | 0.8007 | 0.09227 | 0.009 |
| Valid | 25 | 1.66 | 2.24 | 3.90 | 3.2533 | 0.34047 | 0.116 |

**Table 4.** Statistics.

|  | $\rho_i(HCO_3)$ | $\rho_i(Cl)$ | $\rho_i(SO_4)$ | $\rho_i(Na)$ | Number of Votes |
|---|---|---|---|---|---|
| Valid | 25 | 25 | 25 | 25 | 25 |
| Missing | 0 | 0 | 0 | 0 | 0 |

To solve the problem of comparative classification in an information system in electronic resources based on an algorithm for calculating estimates, a correct algorithm was used

$$A = \left( \sum_{i=1}^{l} \sum_{S^t \in K_j} B(S^t) \cdot C(C_1, C_2) \right) \tag{44}$$

The work of the recognition algorithm in this case is based on the analysis of the data structure and its importance for classification purposes. With this approach, reliability is

assessed by the quality of work of the recognition algorithm on control material. In other words, the algorithm carries out a classification and its results are compared with those that are known a priory. Obviously, with a sufficiently large volume of control, the results reflect the quality of the algorithm. It was this approach that was implied and incorporated into the software package, the application of which to the initial information made it possible to obtain the following: relative weights of feature groups $\rho_i(S, S') = \min p(a_i - b_i)$, vote formula $G_j(S^i) = \frac{1}{\rho_j(S,S')+1}$ and the sum of operators $B_1 + B_2 + B_3 + B_4 = G_1 + G_2 + G_3 + G_4$.

| Salinity types | $a_i(HCO_3)$ | $a_i(Cl)$ | $a_i(SO_4)$ | $a_i(Na)$ | $b_i(HCO_3)$ | $b_i(Cl)$ | $b_i(SO_4)$ | $b_i(Na)$ | $\rho_i(S_i,\tilde{S})=\min\rho_i(|a_i-b_i|)$ | | | | $G_i(S_i,\tilde{S})=\frac{1}{\rho_i(S_i,\tilde{S})+1}$ | | | | $B_1+B_2+B_3+B_4=|G_1+G_2+G_3+G_4|$ |
|---|---|---|---|---|---|---|---|---|---|---|---|---|---|---|---|---|---|
| Non-saline | 0.002 | 0.001 | 0.028 | 0.001 | | | | | 0.07 | 0.499 | 0.472 | 0.249 | 0.934 | 0.667 | 0.679 | 0.8 | 3.082 |
| | 0.011 | 0.004 | 0.032 | 0.003 | | | | | 0.061 | 0.496 | 0.468 | 0.247 | 0.942 | 0.671 | 0.681 | 0.801 | 3.096 |
| | 0.016 | 0.015 | 0.042 | 0.007 | | | | | 0.056 | 0.485 | 0.458 | 0.243 | 0.946 | 0.675 | 0.685 | 0.804 | 3.113 |
| | 0.018 | 0.02 | 0.046 | 0.008 | | | | | 0.018 | 0.48 | 0.454 | 0.242 | 0.982 | 0.677 | 0.687 | 0.805 | 3.153 |
| | 0.019 | 0.025 | 0.048 | 0.009 | | | | | 0.053 | 0.475 | 0.452 | 0.241 | 0.949 | 0.679 | 0.688 | 0.805 | 3.123 |
| Slightly saline | 0.02 | 0.035 | 0.05 | 0.01 | | | | | 0.052 | 0.465 | 0.45 | 0.24 | 0.95 | 0.68 | 0.689 | 0.806 | 3.126 |
| | 0.022 | 0.045 | 0.054 | 0.014 | | | | | 0.022 | 0.455 | 0.446 | 0.236 | 0.978 | 0.684 | 0.691 | 0.809 | 3.164 |
| | 0.025 | 0.055 | 0.075 | 0.02 | | | | | 0.047 | 0.445 | 0.425 | 0.23 | 0.955 | 0.699 | 0.701 | 0.813 | 3.169 |
| | 0.027 | 0.07 | 0.099 | 0.024 | | | | | 0.045 | 0.43 | 0.401 | 0.226 | 0.956 | 0.704 | 0.713 | 0.815 | 3.19 |
| | 0.029 | 0.09 | 0.11 | 0.028 | | | | | 0.043 | 0.41 | 0.39 | 0.222 | 0.958 | 0.709 | 0.719 | 0.818 | 3.205 |
| Moderately saline | 0.03 | 0.12 | 0.12 | 0.03 | | | | | 0.03 | 0.38 | 0.38 | 0.22 | 0.97 | 0.714 | 0.724 | 0.819 | 3.229 |
| | 0.034 | 0.16 | 0.14 | 0.034 | | | | | 0.038 | 0.34 | 0.36 | 0.216 | 0.963 | 0.735 | 0.735 | 0.822 | 3.256 |
| | 0.038 | 0.2 | 0.2 | 0.041 | 0.072 | 0.5 | 0.5 | 0.25 | 0.034 | 0.3 | 0.3 | 0.209 | 0.967 | 0.757 | 0.769 | 0.827 | 3.321 |
| | 0.042 | 0.24 | 0.26 | 0.058 | | | | | 0.03 | 0.26 | 0.24 | 0.192 | 0.97 | 0.781 | 0.806 | 0.838 | 3.397 |
| | 0.048 | 0.29 | 0.34 | 0.095 | | | | | 0.024 | 0.21 | 0.16 | 0.155 | 0.976 | 0.819 | 0.862 | 0.865 | 3.524 |
| Highly saline | 0.05 | 0.32 | 0.35 | 0.1 | | | | | 0.022 | 0.18 | 0.15 | 0.15 | 0.978 | 0.833 | 0.869 | 0.869 | 3.55 |
| | 0.065 | 0.36 | 0.39 | 0.17 | | | | | 0.007 | 0.14 | 0.11 | 0.08 | 0.993 | 0.877 | 0.9 | 0.925 | 3.697 |
| | 0.075 | 0.46 | 0.48 | 0.29 | | | | | 0.003 | 0.04 | 0.02 | 0.04 | 0.997 | 0.961 | 0.980 | 0.961 | 3.9 |
| | 0.09 | 0.57 | 0.62 | 0.43 | | | | | 0.018 | 0.07 | 0.12 | 0.18 | 0.982 | 1 | 0.892 | 0.847 | 3.722 |
| | 0.095 | 0.59 | 0.69 | 0.48 | | | | | 0.023 | 0.09 | 0.19 | 0.23 | 0.977 | 0.917 | 0.84 | 0.813 | 3.548 |
| Very highly saline | 0.1 | 0.61 | 0.7 | 0.5 | | | | | 0.028 | 0.11 | 0.2 | 0.25 | 0.972 | 0.909 | 0.833 | 0.8 | 3.515 |
| | 0.2 | 0.63 | 0.72 | 0.54 | | | | | 0.128 | 0.13 | 0.22 | 0.29 | 0.886 | 0.892 | 0.819 | 0.775 | 3.374 |
| | 0.35 | 0.64 | 0.83 | 0.72 | | | | | 0.278 | 0.14 | 0.33 | 0.47 | 0.782 | 0.847 | 0.751 | 0.68 | 3.062 |
| | 0.45 | 0.68 | 1.18 | 0.98 | | | | | 0.378 | 0.18 | 0.68 | 0.73 | 0.725 | 0.735 | 0.595 | 0.578 | 2.634 |
| | 0.55 | 1.3 | 1.5 | 1.2 | | | | | 0.478 | 0.8 | 1 | 0.95 | 0.676 | 0.555 | 0.5 | 0.512 | 2.244 |

**Figure 2.** Table of attributes belonging to classes.

Example for *Cl*:

$\rho_1(S, S') = |0.001 - 0.5| = 0.499$    $\rho_2(S, S') = |0.004 - 0.5| = 0.496$

$\rho_3(S, S') = |0.015 - 0.5| = 0.485$    $\rho_4(S, S') = |0.02 - 0.5| = 0.48$

$\rho_5(S, S') = |0.025 - 0.5| = 0.475$    $\rho_6(S, S') = |0.035 - 0.5| = 0.465$

$\rho_7(S, S') = |0.045 - 0.5| = 0.455$    $\rho_8(S, S') = |0.055 - 0.5| = 0.445$

$\rho_9(S, S') = |0.07 - 0.5| = 0.43$    $\rho_{10}(S, S') = |0.09 - 0.5| = 0.41$

$\rho_{11}(S, S') = |0.12 - 0.5| = 0.38$    $\rho_{12}(S, S') = |0.16 - 0.5| = 0.34$

$\rho_{13}(S, S') = |0.2 - 0.5| = 0.3$    $\rho_{14}(S, S') = |0.24 - 0.5| = 0.26$

$\rho_{15}(S, S') = |0.29 - 0.5| = 0.21$    $\rho_{16}(S, S') = |0.32 - 0.5| = 0.18$

$\rho_{17}(S, S') = |0.36 - 0.5| = 0.14$    $\rho_{18}(S, S') = |0.46 - 0.5| = 0.04$

$\rho_{19}(S, S') = |0.57 - 0.5| = 0.07$    $\rho_{20}(S, S') = |0.59 - 0.5| = 0.09$

$\rho_{21}(S, S') = |0.61 - 0.5| = 0.11$    $\rho_{22}(S, S') = |0.63 - 0.5| = 0.13$

$\rho_{23}(S, S') = |0.64 - 0.5| = 0.14$    $\rho_{24}(S, S') = |0.68 - 0.5| = 0.18$

$\rho_{25}(S, S') = |1.3 - 0.5| = 0.8$

$G_1(S, S') = \frac{1}{0.499+1} = 0.6671$    $G_2(S, S') = \frac{1}{0.496+1} = 0.6684$

$G_3(S, S') = \frac{1}{0.485+1} = 0.6734$    $G_4(S, S') = \frac{1}{0.48+1} = 0.67567$

$G_5(S, S') = \frac{1}{0.475+1} = 0.67796$    $G_6(S, S') = \frac{1}{0.465+1} = 0.68259$

$G_7(S, S') = \frac{1}{0.455+1} = 0.68728$    $G_8(S, S') = \frac{1}{0.445+1} = 0.69204$

$G_9(S, S') = \frac{1}{0.43+1} = 0.6993$    $G_{10}(S, S') = \frac{1}{0.41+1} = 0.7092$

$G_{11}(S, S') = \frac{1}{0.38+1} = 0.7246$    $G_{12}(S, S') = \frac{1}{0.34+1} = 0.7462$

$G_{13}(S, S') = \frac{1}{0.3+1} = 0.79365$    $G_{14}(S, S') = \frac{1}{0.26+1} = 0.79365$

$$G_{15}(S,S') = \frac{1}{0.21+1} = 0.8264 \qquad G_{16}(S,S') = \frac{1}{0.18+1} = 0.8474$$
$$G_{17}(S,S') = \frac{1}{0.14+1} = 0.87719 \qquad G_{18}(S,S') = \frac{1}{0.04+1} = 0.9615$$
$$G_{19}(S,S') = \frac{1}{0.07+1} = 0.9345 \qquad G_{20}(S,S') = \frac{1}{0.09+1} = 0.9174$$
$$G_{21}(S,S') = \frac{1}{0.11+1} = 0.9009 \qquad G_{22}(S,S') = \frac{1}{0.13+1} = 0.8849$$
$$G_{23}(S,S') = \frac{1}{0.14+1} = 0.8772 \qquad G_{24}(S,S') = \frac{1}{0.18+1} = 0.84745$$
$$G_{25}(S,S') = \frac{1}{0.6+1} = 0.625$$

## 8. Conclusions

The solution to the problem of identifying objects in the IoT ecosystem of the Aral region was analyzed. The problem of constructing a correct algorithm with linear closure operators of a model for calculating estimates for identifying objects in the IoT ecosystem of the Aral region was considered.

Within the framework of the algebraic approach, several variants of linear combinations of recognition operators were constructed, the use of which gives the correct answer on the control material, and this was proven in the form of theorems.

An operator belonging to the linear closure of the model of the type of calculation of estimates was constructed, which was the sum of q operators of the model of calculation of estimates and was described by a set of numerical parameters $3 \cdot n \cdot m \cdot q$, where $n$ was the number of specified characteristics, $m$ was the number of reference objects, and $q$ was the set of recognized objects. The completeness of the linear closure of this model was proven for all problems in which for each class there is at least one stationary pair $(u, v)$, and this correct algorithm $A$ was written explicitly.

The results obtained in this article, namely the proven theorems, made it possible to construct correct algorithms on control material based on a combination of linear recognition operators.

The constructed correct recognition algorithms, which are the easiest to use, where there is no optimization procedure, made it possible to quickly solve the issues of identifying incoming information flows in the IoT ecosystem of the Aral region.

**Author Contributions:** Conceptualization, A.K. and I.S.; methodology, A.K.; software, I.S.; validation, A.K., I.S. and A.B. (Akbarjon Babadjanov); formal analysis, I.S.; investigation, A.K.; resources, I.S. and A.B. (Alimdzhan Babadzhanov); data curation, I.S.; writing—original draft preparation, A.K., I.S. and A.B. (Akbarjon Babadjanov); writing—review and editing, A.K., I.S., A.B. (Akbarjon Babadjanov) and A.B. (Alimdzhan Babadzhanov); visualization, I.S.; supervision, A.K. All authors have read and agreed to the published version of the manuscript.

**Funding:** This research received no external funding.

**Data Availability Statement:** No new data generated. Data sharing not applicable to this article.

**Conflicts of Interest:** The authors declare no conflicts of interest.

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
