# Peer review of "Algebraic Recognition Approach in IoT Ecosystem"

_mathematics, doi:10.3390/math12071086_

Round 1

Reviewer 1 Report

Comments and Suggestions for Authors

First of all, I note the work done by the authors for the creation of this article. Unfortunately, there are more aspects of form that authors must consider in order to create a publishable article.

It would be good if the title of the article is better correlated with the content. The use of "IoT" in the title and then only in the keywords and references, without even a mention in the text, can generate confusion. The applicability of the presented recognition algorithm is considered only in the results paragraph (line 631-654). Here we present the data taken from the sensors regarding the chemical composition (their classification).

Notations are used without much explanation, such as "3 · n · m · q". Only q is explained, otherwise this notation (line 4, line 73, line 657) is seen as "numerical parameters".

At line 78, I think is a typo ("etc." must be deleted). There are some relation unnumbered (e.g. p. 3, between line 100 and line 101, there are no numbering even text). Near line 100 is a mistake using letter "C" (used fordecision rule) instead "c" (real number). In Table 1, what is the meaning of "Y/X" ?

Authors must be more consistent in writing. If it provides proof for most of the theorems/lemmas, there is still a different treatment for Theorem 1(line 225) and lemma 2 (line 390). There are no related proofs.

It would be good if the article was rewritten in a more succinct style, with a greater capacity for synthesis on the part of the authors, but without losing the information presented.

Author Response

It would be good if the title of the article is better correlated with the content. The use of "IoT" in the title and then only in the keywords are references, without even a mention in the text, can generate confusion. The applicability of the presented recognition algorithms is considered only in the results paragraph (line 631-654). Here we present the data taken from the sensors regarding the chemical composition (their classification).

We agree to remove the word IoT from the name. The title of the Article will be: Algebraic recognition approach in ecosystem

To solve the problem of comparative classification in an information system in electronic resources based on an algorithm for calculating estimates, a correct algorithm was used.

The work of the recognition algorithm in this case is based on the analysis of the data structure and its importance for classification purposes. With this approach, reliability is assessed by the quality of work of the recognition algorithm on control material. In other words, the algorithm carries out a classification and its results are compared with those that are known a priori. Obviously, with a sufficiently large volume of control, the results well reflect the quality of the algorithm. It was this approach that was implied and incorporated into the software package, the application of which to the initial information made it possible to obtain the following: relative weights of feature groups , vote formula  and the sum of operators

Example for .:

Fig 1. Table of attributes belonging to classes

Since Alimdzhan Babadzhanov obtained the result of the experiment, we include him among the co-authors

Notations are used without much explanation, such as "3*n*m*q". Only "q" is explained, otherwise this notation (line 4, line 73, line 657) is seen as "numerical parameters

where n is the number of predetermined features, m is the number of reference objects, q is the set of recognizable objects

At line 78, I think is a typo ("etc." must be deleted). There are some relation unnumbered (e.g. p.3, between line 100 and line 101, there are no numbering even text). Near line 100 is a mistake using letter "C" (used fordecision rule) instead "c" (real number). In Table 1, what is meaning of "Y/X"?

yes it was a typo, we removed the “etc.” We replaced the letter "c". Correct presentation of the table. 1 next. Y is the designation of columns, X is the designation of rows.

Authors must be more consistent in writing. If it provides proof for most of the theorems/lemmas, there is still a different treatment for Theorem 1 (line 225) and lemma (line 390). There are no related proofs.

Theorem 1 is a well-known theorem, so we provide a link to the article [Zhuravlev].

Lemma 2 is a consequence of Lemma 1, so it needs to be replaced by a consequence.

It would be good if the article was rewritten in a more succinct style, with a greater capacity for synthesis on the part of the authors, but without losing the information presented.

The article is written in a good scientific style, we think this does not need to be shortened

Reviewer 2 Report

Comments and Suggestions for Authors
  • The abstract and conclusions lack specific results, including critical findings
  • What is the objectives of the research and novelty of the research, highlight about it.
  • In the title it is mentioned IoT ecosystem but in abstract it is not discussed
  • All equations must be numbered and referred in the text
  • Why authors selected nearest neighbor algorithm and average distance algorithm
  • Results of the research need to be compared with previously carried out research and how this is performance is better or not to be discussed.
  • In the entire manuscript nowhere IoT is correlated with algorithms used in this research. Hence authors required to discuss in details about the applicability IoT with algorithms.
  • Discuss about the societal benefits of this research outcomes
  • The paper lacks details on the methodology used to develop and validate the algebraic recognition approach.
  • The writing style is verbose and difficult to parse, hindering understanding.
Comments on the Quality of English Language
  • The abstract and conclusions lack specific results, including critical findings
  • What is the objectives of the research and novelty of the research, highlight about it.
  • In the title it is mentioned IoT ecosystem but in abstract it is not discussed
  • All equations must be numbered and referred in the text
  • Why authors selected nearest neighbor algorithm and average distance algorithm
  • Results of the research need to be compared with previously carried out research and how this is performance is better or not to be discussed.
  • In the entire manuscript nowhere IoT is correlated with algorithms used in this research. Hence authors required to discuss in details about the applicability IoT with algorithms.
  • Discuss about the societal benefits of this research outcomes
  • The paper lacks details on the methodology used to develop and validate the algebraic recognition approach.
  • The writing style is verbose and difficult to parse, hindering understanding.

Author Response

  • What is the objectives of the research and novelty of the research, highlight about it.

The purpose of the study is to develop effective object recognition methods to solve the identification problem in the IoT ecosystem of the Aral region

The novelty of the research lies in the development of correct recognition algorithms based on an algebraic approach in solving the problem of identifying the flow of information in the IoT ecosystem of the Aral region.

  • In the title it is mentioned IoT ecosystem but in abstract it is not discussed

The ecosystem is described in detail in the works [Saymanov, I.; Logical recognition method for solving the problem of identification in the Internet of Things. arxiv 2024, arXiv:2402.04338.] of the authors of this work. It covers a network of sensors for determining the level of groundwater and salinity of water and soil and transmitting information via communication channels to the System Server for identification and further processing.

  • All equations must be numbered and referred in the text

ok, let's take it into account

  • Why authors selected nearest neighbor algorithm and average distance algorithm

The nearest neighbor algorithm and the average distance algorithm are simpler algorithms in complexity compared to other recognition algorithms.

The results obtained in the article, namely the proven theorems, make it possible to construct a correct algorithm on control material based on a combination of linear recognition operators. And this allows you to interactively solve the issues of identifying incoming information flows in the IoT ecosystem.

  • Results of the research need to be compared with previously carried out research and how this is performance is better or not to be discussed.

Classical pattern recognition algorithms, for example algorithms for calculating scores, solve a complex optimization problem in order to achieve high percentages of recognition on control material. In the work, based on the proof of theorems, correct algorithms were constructed based on a linear combination of recognition operators, which are the easiest to use, where there is no optimization procedure.

  • In the entire manuscript nowhere IoT is correlated with algorithms used in this research. Hence authors required to discuss in details about the applicability IoT with algorithms.

The article solves the applied problem of identification in the ecosystem of the Aral region, where formulas for recognizing correct algorithms are specifically given. Moreover, the correctness is proven in the theorems of this work in terms of the algebraic approach.

  • Discuss about the societal benefits of this research outcomes

The public benefit of the results of this study is that the results are used in the ecosystem of the Aral region in monitoring the use of water resources and calculating salinity levels for agricultural use in an interactive mode.

  • The paper lacks details on the methodology used to develop and validate the algebraic recognition approach.

The methodology used to develop and test the algebraic recognition approach was proposed and described in the works of academician Yu.I. Zhuravlev and his students. This work is based on the ideas of the algebraic approach of Academician Yu.I. Zhuravlev.

Round 2

Reviewer 1 Report

Comments and Suggestions for Authors

Now the authors presented the applicability more clearly. It seems that there are no more problems regarding the content of the article. All my observations were taken into account and the authors corrected or modified, if necessary.